# Appearances Can Be Deceptive: Morphological, Phylogenetic, and Nomenclatural Delineation of Two Newly Named African Species Related to *Frankenia pulverulenta* (Frankeniaceae)

**DOI:** 10.3390/plants14071130

**Published:** 2025-04-05

**Authors:** María Ángeles Alonso, Manuel B. Crespo, Jordi Abad-Brotons, Mario Martínez-Azorín, José Luis Villar

**Affiliations:** Departamento de Ciencias Ambientales y Recursos Naturales (dCARN), Universidad de Alicante, P.O. Box 99, ES-03080 Alicante, Spain; crespo@ua.es (M.B.C.); jordi.abad@ua.es (J.A.-B.); mmartinez@ua.es (M.M.-A.); jose.villar@ua.es (J.L.V.)

**Keywords:** African flora, *Frankenia*, Frankeniaceae, ITS, *matK*, new names, nomenclature, phylogenetic analyses, taxonomy

## Abstract

*Frankenia* is a morphologically complex genus, with some species exhibiting a few diagnostic characters and significant morphological variability. This has led to misidentification or the synonymisation of many names based on one or a few diagnostic traits. This phenomenon affects the annual sea-heath, *F. pulverulenta*, a Eurasian–Mediterranean herb that has become subcosmopolitan, to which several entities have been included due to their shared features, namely their annual lifespan or their flattened leaves. However, this fact also extends to shrubby species, such as the Madeiran *F. cespitosa*. Here, integrative taxonomic studies, encompassing detailed morphological descriptions of macro- and microcharacters along with molecular phylogenetic analyses of both nuclear ribosomal (ITS1-5.8S-ITS2 region) and plastid (*matK* gene) DNA sequence data, and an analysis of biogeographic data were undertaken. This examination has resulted in the most complete phylogenetic trees of *Frankenia* to date, leading to the reinstatement of two African species broadly differing morphologically from *F. pulverulenta*. Firstly, *F. florida* L.Chevall., a name applied to a species occurring in the Saharan regions of Algeria, Morocco, Mali, and Mauritania, is often accepted as a variety or subspecies of the annual sea-heath. In contrast, *F. densa* Pohnert, a species endemic to southern Namibia and northern South Africa, has been synonymised with *F. pulverulenta*. However, since those two names were later homonyms of two Chilean and Australian plants, they were deemed illegitimate upon publication. Consequently, two new names are proposed for them: *F. sahariensis* and *F. dinteri*, respectively. Their substantiation as independent species is provided by data on their morphology, distribution, ecology, and molecular phylogenetics, which demonstrate their distinctiveness from *F. pulverulenta*. Nomenclatural synonymy and types are also presented for all concerned names, including the designation of two new lectotypes. Furthermore, the importance of an accurate description of the morphological variation in populations is emphasised for a precise identification of taxa in *Frankenia*.

## 1. Introduction

The genus *Frankenia* L. is the sole member currently recognised within Frankeniaceae Desv., *nom. cons.*, which has been expanded to include *Anthobryum* Phil., *Beatsonia* Roxb., *Hypericopsis* Boiss. and *Niederleinia* Hieron. ([1,2,3,4], and references therein). Broadly speaking, about 80–90 species are often accepted as part of this family, which are distributed in temperate or tropical regions, commonly in Mediterranean-type areas around the world, but with ca. 40 species found in Australia [5,6]. Most members of *Frankenia* exhibit a woody or suffruticose habit, erect to prostrate, with sometimes densely pulvinate or with subspinescent branches, but some others are prostrate to ascendant annual herbs [7,8]. With very few exceptions, most species usually grow in saline soils, including in coastal and inland salt marshes and gypsum- or calcium-rich outcrops [9], often enduring semi-arid to arid climates that mostly show prevalent winter rainfall [10]. One such exception is the South African narrow endemic *F. fruticosa* J.C. Manning & Helme, which occurs on quartz soils [11]. Because most taxa grow under strongly stressing conditions, many endemics are found that are restricted to small areas or archipelagos [12], whereas others, such as the annual sea-heath, *F. pulverulenta* L., have become subcosmopolitan [13,14]. These peculiar lifestyles have led to the acquisition of specialised structures, such as salt-secreting multicellular glands that often generate whitish salt crusts on leaves and stems [8]. This characteristic is shared with representatives of some related families within the Caryophyllales order, such as *Plumbaginaceae* Juss., *nom. cons.*, or Tamaricaceae Link, *nom. cons.* [15], although in each case, they exhibit distinct structural features [16].

From a taxonomic perspective, the first comprehensive treatment of the genus was undertaken by Niedenzu [7], who accepted *Frankenia*, *Anthobryum*, *Beatsonia*, *Hypericopsis* and *Niederleinia* as distinct genera. Just a few years later, Summerhayes [5] published a revision of the Australian taxa and described most of the currently accepted species. More recently, Whalen [17] conducted a taxonomic study on the American Frankeniaceae, whereby a detailed morphological revision was presented along with a discussion on the status of the South American genera *Anthobryum* and *Niederleinia*, which she synonymised with *Frankenia*. However, while being a good starting point for a treatment of the whole genus, Niedenzu’s [7] account is incomplete and requires a thorough update. Among the partial monographs, different authors have presented taxonomic arrangements for the genus in large regions of Eurasia and Africa. After revisions for Flora of the USSR [18], Flora europaea [19] and Flora iranica [20], with some additions by Lomonosova [21], up to seven species plus twelve infraspecific taxa (subspecies, varieties, and forms) are now widely accepted.

For Africa, the revision of the Moroccan taxa by Nègre [22] stands out, in which the author studied in detail the circumscription and synonymy of taxa previously summarised by Jahandiez and Maire [23] and Emberger and Maire [24] for Morocco and its neighbouring areas. He accepted four species plus fourteen subspecific and varietal taxa there. More recently, Crespo et al. [12] clarified the circumscription of some neglected names concerning southern African plants and recognised seven species, most of them endemics often misidentified with other widespread relatives. However, the diverse taxonomic criteria adopted within the last few decades by authors dealing with *Frankenia* in diverse territories have generated quite heterogeneous treatments that makes a global understanding of the genus much more difficult for most taxonomists.

Furthermore, molecular studies that include species of *Frankenia* are also scarce. In a work focused on the relationships between Frankeniaceae and Tamaricaceae based on nuclear ribosomal DNA (nrDNA) sequences, Gaskin et al. [3] included eight *Frankenia* species from North America, Australia, and Eurasia, such as the outstanding *F. persica* (Boiss.) Jaub. & Spach (≡*Hypericopsis persica* Boiss.). The most recent phylogenetic studies on the genus were published by Crespo et al. [12], who presented a preliminary tree based on internal transcribed spacer (ITS) sequences, including from 16 species from Eurasia and Africa, which served as a basis to recover or describe some neglected species from South Africa often misidentified with *F. pulverulenta*.

Very few species in this genus are defined as annual or biennial. In particular, the annual sea-heath, *Frankenia pulverulenta*, is a Eurasian–Mediterranean ephemeral herb that today is broadly distributed around the world [14], occurring in saline soils disturbed by human activities [13,14,15,16,17]. According to Crespo et al. [12], it is typically a prostrate plant with many-branched non-rooting stems that are densely and minutely pubescent on one side; leaves that are flat, broadly obovate–cuneate, subconcolorous, glabrous on the adaxial surface and variably hairy on the abaxial surface, rounded and emarginate at the apex and cuneate at the base; pentamerous flowers that are small, axillary, solitary, and scattered; calyx up to 5 mm long; pinkish corolla up to 5 mm; a 3-carpelar capsule; three placentas, with 12–20 ovules each; numerous small seeds (up to 0.5–0.7 mm long) that are ellipsoid, monosulcate, have weakly and irregularly ornamented testa, are very sparsely covered with small-sized papillae (4–17 µm), and homogeneous (conical–obtuse) in shape.

Recent investigations conducted in southern Africa, through both fieldwork and herbarium studies [12], revealed the existence of peculiar populations showing some superficial resemblance to *F. pulverulenta*, mostly on account of their flattened roundish to broadly ovate–oblong leaves. However, those authors demonstrated that several distinct biological entities had been overlooked under that designation within the last decades. On the one hand, the name *F. nodiflora* Lam., often synonymised with *F. pulverulenta*, was restored for littoral populations of creeping prostrate shrublets with concolorous glabrous leaves and dense glomerular inflorescences, occurring in saltmarshes and saltpans of Cape Flats (Western Cape Province). On the other hand, *F. nummularia* M.B.Crespo, M.Á.Alonso, Mart.-Azorín, J.L.Villar & Mucina was newly described for populations of creeping prostrate shrublets with discolorous glabrous leaves and loosely dichasial, usually widely branched inflorescences, found in inland salt pans and saline river beds in the southwestern part of the Karoo Region (Northern and Western Cape Provinces). These had often been misidentified as the annual sea-heath, *F. pulverulenta*.

Similar research on other African populations of annual or short-lived perennial plants often related to *F. pulverulenta* also unveiled surprising data. Firstly, populations of a North African plant occurring in saline substrates of the western Saharan regions were described as *F. florida* L.Chevall., a later homonym of the Chilean *F. florida* Phil. and, hence, an illegitimate name, according to the *Shenzhen Code* (hereafter ICN; [25]). That North African name has often been treated as *F. pulverulenta* var. *florida* Maire or subsp. *florida* (Maire) Maire [4,26] or merely as a synonym of the annual sea-heath. Secondly, populations of a remarkable species growing in saline soils of inland Namibia were described as *F. densa* Pohnert, a later homonym of the Australian *F. densa* Summerh., which is, hence, illegitimate and unavailable for use. However, despite several characteristics that allow its separation from the morphologically close *F. pulverulenta*, Pohnert’s name is currently reduced to a synonym of the latter [4].

In the present contribution, as part of our ongoing integrative taxonomic studies on the genus *Frankenia*, the morphology and phylogenetic relationships of both *F. florida* L.Chevall. and *F. densa* Pohnert are studied. The primary objective of this study was to compare both taxonomic entities to the widespread *F. pulverulenta*, with the aim of identifying their morphological and molecular similarities and divergences, which would help us decide the most appropriate taxonomic treatment for each of them. On that basis, morphological, distributional, ecological, and molecular phylogenetic data are shown that support the recognition of both entities as independent species that are not related to *F. pulverulenta*, for which the new names *F. sahariensis* and *F. dinteri* are respectively proposed. Nomenclatural synonyms and types are also reported for all concerned names, including the designation of two new lectotypes.

## 2. Materials and Methods

### 2.1. Morphological and Habitat Studies

Detailed morphological observations were conducted on both living plants from wild populations and dried herbarium specimens sourced from the herbaria ABH, BM, BOL, G-DC, K, LINN, MA, MPU, NBG, P, and W (acronyms according to Thiers [27]), using an OLYMPUS SZX7 (Olympus Europa, Hamburg, Germany) binocular microscope. Digital sheets from the herbaria GH, GZU, HBG, JE, LY, M, MO, PRE, RSA, US, WAG, and Z were reviewed as well. A list of the herbarium material examined, including nomenclatural types, is reported for each species, with barcode numbers provided after each herbarium acronym where available. In summary, 54 herbarium vouchers were examined for *F. sahariensis*, 32 for *F. dinteri*, and 47 for *F. pulverulenta* (Appendix B), which provide a comprehensive picture of the morphological variability of these species in their respective native distribution areas.

Scanning electron microscope (SEM) micrographs of seeds were captured using a JEOL JSM-IT500HR (JEOL Ltd., Tokyo, Japan) operating at 15 kV. No special treatment was needed for the materials before observation. A minimum of 10 mature seeds from different individuals of each studied species (Table 1) were carefully examined. The samples were affixed to metallic stubs and coated with 10 nm of platinum in a QUORUM Q150T ES Plus (Quorum Technologies, Laughton, UK) sputter coater. Measurements on the SEM micrographs were conducted using ImageJ v.1.8.0 [28].

The authors of the taxa cited follow the guidelines of IPNI [29]. Nomenclatural issues accord with the *Shenzhen Code* (ICN; [25]). For the new species from southern Africa, the orthography of geographical names and the grid-number system follow Leistner and Morris [30], as defined by the South African topographical map sheet referencing system [31]; this is broadly known as a ”quarter-degree square” (QDS) grid (a brief explanation on how it works is shown in Crespo et al. [12]).

The bioclimatic and biogeographic classification of the southern African species is based on Mucina and Rutherford [32]. For North African species, the biogeographical characterisations of Quézel [33] and Takhtajan [34] were also considered. Distribution maps were created using cartography tools from Google Earth.

### 2.2. Molecular Analyses

Silica-gel-dried materials and herbarium vouchers were utilised for total DNA extraction following a modified 2× cetyltrimethylammonium bromide (CTAB) protocol [35]. When herbarium material was unavailable for sampling, silica-gel-dried material from wild populations (one sample per population) of each taxon was used, since the phylogenetic trees remained unchanged after the addition of new samples from the same population. Total DNA was then purified using MoBio mini-columns (MoBio Laboratoires, Carlsbad, CA, USA) and typically stored in 0.1× TE buffer (10 mM Tris-HCl, 1 mM ethylenediaminetetraacetic acid (EDTA), pH 8.0). The internal transcribed spacer (ITS) region (comprising the ITS1 spacer, the 5.8S gene, and the ITS2 spacer) of nuclear ribosomal DNA (nrDNA) was amplified using the ITS5 and ITS4 primers [36], while the Maturase K (*matK*) gene of plastid DNA (cpDNA) was amplified using the XF (TAATTTACGATCAATTCATTC) and 5R (GTTCTAGCACAAGAAAGTCG) primers [37]. Amplifications were conducted in a total reaction volume of 25.5 μL, which included 12.5 μL of DreamTaq PCR Master Mix (2×) (Thermo Scientific, Waltham, MA, USA), 10 μL of distilled water, 0.5 μL of 0.4% bovine serum albumin (BSA), 0.5 μL of dimethyl sulfoxide (DMSO), 0.5 μL of each primer (10 pmol/μL), and 1 μL of template DNA, all processed on a 9700 GeneAmp (Applied Biosystems, Foster City, CA, USA) thermocycler. The PCR program for the ITS involved an initial step of 2 min at 95 °C, followed by 35 cycles of 95 °C for 1 min, 53 °C for 1 min, and 72 °C for 2 min, concluding with a final extension at 72 °C for 5 min. The PCR program for *matK* included an initial step of 2 min at 94 °C, followed by 32 cycles of 94 °C for 1 min, 53 °C for 1 min, and 72 °C for 1 min and 30 s, finishing with a final extension at 72 °C for 4 min.

Sanger sequencing was performed by automated standard techniques. Sequencer 4.1 (Gene Codes Corp., Ann Arbor, MI, USA) was utilised to assemble complementary strands and to verify the software base-calling. Sequence alignment was conducted using MUSCLE [38] within MEGA X v.10.2.6 [39], supplemented by minor manual adjustments to create the final aligned matrix. The analyses incorporated fifty-six samples representing twenty species of *Frankenia*, encompassing the primary Eurasian and African groups within the genus. Members of the sister family Tamaricaceae [1,40], namely *Myricaria germanica* L., *Reaumuria alternifolia* (Labill.) Britten, *R. songarica* (Pall.) Maxim., and *Tamarix gallica* L., were included as outgroups based on previous works [3,12].

Two datasets were constructed: one for the ITS region (including 54 *Frankenia* sequences plus 4 outgroup sequences, totalling 720 positions) and another for the *matK* gene (comprising 38 *Frankenia* sequences plus 3 outgroup sequences, with a total of 829 positions). Specifically for this study, a total of 38 *matK* plus 28 ITS sequences were generated (Table 2), while the remaining ones were sourced from various plant databases according to their GenBank (https://www.ncbi.nlm.nih.gov/genbank/; accessed on 31 January 2025) availability (see [12]). As a result, only a few accessions in the ITS and *matK* datasets originate from different plant sources. Accessions listed in GenBank as “*Reaumuria hypericoides* Willd.” are represented in our trees as *R. alternifolia*, given that the former name is a superfluous synonym for the latter and, as such, has priority. Likewise, the ITS accession OR183467, previously filed as “*Frankenia hirsuta* L.” (voucher ABH-45933!), corresponds to the closely related species *F. salsuginea* Adıgüzel & Aytaç (Table 2).

Phylogenetic analyses of both the ITS region and the *matK* gene were conducted using Maximum Parsimony (MP), Maximum Likelihood (ML), and Neighbour Joining (NJ) methods. The MP analysis was performed on 10,000 replicates using both PAUP, with heuristic search options based on the Nearest Neighbour Interchange (NNI) strategy, and MEGA, with heuristic search options applying the Subtree–Pruning–Regrafting (SPR) method with search level 1 [41] for comparative results. ML [42] and NJ [43] analyses were also executed in MEGA, alongside the selection of the optimal model of DNA substitutions for each method. Models with the lowest Bayesian Information Criterion (BIC) scores were deemed most suitable for describing the substitution patterns in the ML and NJ analyses. Additionally, jModelTest 2.1.10 [44] was utilised to identify the best model of DNA substitutions for the Bayesian Inference (BI) analyses, using the Akaike Information Criterion (AIC, [45]). Phylogenetic reconstructions for Maximum Likelihood (ML) and evolutionary distances for the Neighbour Joining (NJ) method were estimated using the Kimura 2-parameter (K2) model [46] for the ITS matrix. This model allowed for some sites to be evolutionarily invariable, with 26.46% of sites fitting this criterion (+*I*). In contrast, the *matK* matrix utilised the Tamura 3-parameter (T92) model [47] coupled with a discrete Gamma distribution (*G* = 1.0742) to model the rate variation among sites. For these analyses, all sites in the matrices were considered. To ensure comparability, ambiguous positions were removed for each sequence pair using the pairwise deletion option. This removal did not result in significant differences in the obtained phylogenies, although it affected support percentages for a few branches. Support for all the methods was assessed using bootstrapping [48] with 10,000 replicates, retaining only 10 trees per replicate. Clades with bootstrap percentage (BP) values of 50–74% were considered weakly supported, while values of 75–89% were considered moderately supported, and those of 90–100% strongly supported.

Furthermore, Bayesian inference (BI) analyses were conducted using MrBayes 3.2 [49], where the Markov Chain Monte Carlo (MCMC) algorithm was executed for 10 million generations, with samples taken every 1000 generations. Two independent runs were carried out. The analyses employed the General Time Reversible (GTR) + proportion of invariant sites (*I*) + Gamma distribution (*G*) model, as suggested by the results from jModelTest based on the Akaike Information Criterion (AIC). The initial 25% of generations were excluded (burninfrac = 0.25), and the remaining trees were utilised to compile a posterior probability (PP) distribution through a 50% majority-rule consensus. Clade support was again evaluated using bootstrap analysis [48] with 10,000 replicates while retaining only 10 trees per replicate. Clades with PP ≥ 0.95 were regarded as strongly supported.

**Table 2 plants-14-01130-t002:** List of outgroups and *Frankenia* accessions used for the phylogenetic analyses, with data on provenance and source plus Genbank codes of the ITS and *matK* accessions utilised.

Taxon	Provenance (Herbarium Voucher)	Source	ITS	Source	*matK*
*Tamarix gallica* L.	France: Saintes Maries de la Mer (ABH-57865)	Villar et al. [50]	MH626294	-	-
	Italy: Sardinia	Meimberg et al. [40]	-	-	AF204861
*Myricaria germanica* L.	Kazakhstan: Zajsanskaya depression (LE)	Zhang et al. [51]	KJ808607	-	-
	China: Unspecified (CPG-11863)	Chen et al. [52]	-	-	KX526795
*Reaumuria alternifolia* (Labill.) Britten ^1^	Azerbaijan: Caucasus (MW)	Zhang et al. [53]	KJ729627	-	-
*Reaumuria songarica* (Pall.) Maxim.	China: Xinjiang (M)	Zhang et al. [51]	OQ617495	-	-
	China: Xinjiang (TURP)	Song et al. [54]	-	-	MT918094
*Frankenia anneliseae* M.B.Crespo & al.	South Africa: Klipfontein (ABH-76891)	Crespo et al. [12]	OR183455	-	-
	South Africa: Skoverfontein (ABH-83196)	Crespo et al. [12]	OR183456	This paper	PV258653
	South Africa: Klipfontein (ABH-76872)	-	-	This paper	PV258654
*F. boissieri* Reut. ex Boiss.	Spain: Huelva, Lepe, El Terrón (ABH-83544)	This paper	PV241633	This paper	PV258655
	Portugal: Algarve, Vale de Parra (ABH-73553)	This paper	PV241634	-	-
*F. cespitosa* Lowe	Portugal: Madeira Is., Porto Santo, Morenos (MA-902612)	This paper	PV241635	-	-
*F. capitata* Webb & Berthel.	Spain: Canary Is., Las Palmas de Gran Canaria, Isleta (ABH-83612)	Crespo et al. [12]	OR183458	This paper	PV258656
	Spain: Canary Is., Lanzarote, Teguise (ABH-83884)	This paper	PV241636	-	-
	Spain: Canary Is., Lanzarote, Yaiza (ABH-83881)	This paper	PV241637	This paper	PV258658
	Spain: Canary Is., Lanzarote, Yaiza (ABH-83880)	-	-	This paper	PV258657
*F. composita* Pau & Font Quer	Morocco: Al Hoceïma, Cala Iris (ABH-81590)	Crespo et al. [12]	OR183459	This paper	PV258659
	Spain: Murcia, El Carmolí (ABH-84195)	This paper	PV241638	This paper	PV258660
*F. corymbosa* Desf.	Spain: Alicante, Santa Pola (ABH-79956)	Crespo et al. [12]	OR183462	This paper	PV258663
	Morocco: Nador, Punta Charrana (ABH-54294)	Crespo et al. [12]	OR183461	This paper	PV258662
	Morocco: Al-Hoceïma (ABH-54526)	Crespo et al. [12]	OR183460	-	-
	Spain: Murcia, Cabo Cope (ABH-83531)	Crespo et al. [12]	OR183463	This paper	PV258661
*F. dinteri* M.Á.Alonso & al., nom. nov.	Namibia: Goageb (ABH-76804)	This paper	PV241642	This paper	PV258666
	South Africa: Onseepkans (ABH-83234)	This paper	PV241639	This paper	PV258667
	South Africa: Daberas Farm (ABH-83264)	This paper	PV241640	This paper	PV258665
	South Africa: Augrabies (ABH-83265)	This paper	PV241641	This paper	PV258664
*F. ericifolia* C.Sm. ex DC., nom. cons. prop.	Spain: Canary Isl., Tenerife (ABH-79975)	Crespo et al. [12]	OR183464	This paper	PV258668
	Spain: Canary Isl., Tenerife, Güímar (ABH-83613)	Crespo et al. [12]	OR183465	-	-
	Spain: Canary Isl., Tenerife, Granadilla de Abona (ABH-83873)	This paper	PV241646	This paper	PV258669
	Spain: Canary Isl., Tenerife, Malpaso, *IA3069* (MA)	This paper	PV241643	This paper	PV258670
	Spain: Canary Isl., Lanzarote, Caleta del Mojón Blanco (ABH-83889)	This paper	PV241644	This paper	PV258671
	Spain: Canary Isl., Lanzarote, Risco de Famara (ABH-83893)	This paper	PV241645	This paper	PV258672
*F. fruticosa* J.C.Mannig & Helme	South Africa: Moedverloren (ABH-76898)	Crespo et al. [12]	OR183466	-	-
*F. hirsuta* L.	Italy: Puglia, Bari (ABH-84234)	This paper	PV241648	This paper	PV258673
	Cyprus: Akamas (MA-526424)	This paper	PV241647	-	-
*F. ifniensis* Caball.	Morocco: Sidi Ifni to Oued Noun (MA-758515)	Crespo et al. [12]	OR183468	This paper	PV258674
	Morocco: El Farsia (MA-712824)	This paper	PV241649	-	-
	Morocco: Gelmim (MA-902279)	This paper	PV241650	-	-
*F. laevis* L.	Libya: Cyrenaica, Jbel Akhdar (MA-826355)	This paper	PV241652	-	-
	France: Aude, Étang de La Palme (ABH-70584)	Crespo et al. [12]	OR183469	This paper	PV258676
	Spain: Mallorca, Conejera (ABH-57810)	This paper	PV241651	This paper	PV258675
*F. nummularia* M.B.Crespo & al.	South Africa: Kookfontein River (ABH-83290)	Crespo et al. [12]	OR183471	-	-
	South Africa: Tankwa Karoo (ABH-83295)	Crespo et al. [12]	OR183472	This paper	PV258677
*F. pseudoericifolia* Rivas Mart. & al.	Portugal: Cape Verde, São Antão (MA-0906845)	Crespo et al. [12]	OR183473	-	-
*F. pulverulenta* L.	South Africa: Redelinghuis (ABH-77205)	Crespo et al. [12]	OR183474	-	-
	South Africa: Skoverfontein (ABH-83195)	This paper	PV241657	-	-
	Spain: Teruel, Alcañiz (ABH-73564)	Crespo et al. [12]	OR183475	-	-
	Spain: Alicante, Cabo de las Huertas (ABH-74763)	This paper	PV241653	This paper	PV258678
	Spain: Albacete, Fuentealbilla (ABH-40820)	This paper	PV241654	-	-
	Morocco: Melga-el-Ouidane (ABH-59986)	This paper	PV241655	This paper	PV258679
	Spain: Canary Isl., Tenerife, Puerto de la Cruz (ABH-79974)	Crespo et al. [12]	OR183477	This paper	PV258680
	Italy: Puglia, Torre Spechiola (ABH-84244)	This paper	PV241656	This paper	PV258681
	Spain: Alicante, Cabo de las Huertas (ABH-41853)	Crespo et al. [12]	OR183476	-	-
*F. repens* (P.J.Bergius) Fourc.	South Africa: S of Hondeklipbaai (ABH-76862)	Crespo et al. [12]	OR183479	-	-
	South Africa: Velddrift (ABH-76849)	-	-	This paper	PV258682
	South Africa: S of Groenrivier (ABH-76868)	Crespo et al. [12]	OR183478	This paper	PV258683
*F. sahariensis* M.Á.Alonso & al., nom. nov.	Morocco: Guelmim to Tan Tan (MA-786164)	This paper	PV241658	This paper	PV258684
	Morocco: Sidi Ifni (MA-913227)	This paper	PV241659	This paper	PV258685
*F. salsuginea* Adıgüzel & Aytaç ^2^	Turkyie: Tuz Gölii, salty lagoon (ABH-45933)	Crespo et al. [12]	OR183467	This paper	PV258687
	Turkyie: Dörtyol (MA-561567)	This paper	PV241660	This paper	PV258686
*F. thymifolia* Desf.	Algeria: Bougtob, Chott Cherguí (ABH-59344)	Crespo et al. [12]	OR183481	This paper	PV258688
	Spain: Zaragoza: Bujaraloz (ABH-75454)	Crespo et al. [12]	OR183480	This paper	PV258689
*F. velutina* Brouss. ex DC.	Morocco: Essaouira (ABH-79929)	Crespo et al. [12]	OR183482	This paper	PV258690

^1^ Filed in GenBank as the synonym name *R. hypericoides* Willd. (nom. illeg.); ^2^ Filed in GenBank as *F. hirsuta* (see Crespo et al. [12]).

## 3. Results

### 3.1. Taxonomic Treatment and Description of New Taxa

#### 3.1.1. ***Frankenia sahariensis*** M.Á.Alonso, M.B.Crespo, Abad-Brotons, Mart.-Azorín & J.L.Villar, ***nom. nov.***

≡*Frankenia florida* L.Chevall. in Bull. Herb. Boissier ser. 2, 3(9): 768. 1903 [replaced synonym], *nom. illeg.* [*non* Phil. in Anales Univ. Chile 41: 676. 1872] ≡ *F. pulverulenta* var. *florida* Maire in Bull. Soc. Hist. Nat. Afrique N. 27: 210. 1936 ≡ *F. pulverulenta* subsp. *florida* (Maire) Maire, Cat. Pl. Maroc 4: 1071. 1941. *Type*: Algeria. [El Menia Province:] El Goléa [currently El Menia], in arenosis salsis. April 1902, *L. Chevallier 404* (**lectotype, designated here**: P-06618528!, Figure 1; isolectotypes: P-05145114!, P-06618529!, P-06618525!, MPU-007119!, MPU-007120!, US-00679979 [digital image!], GZU-000269792 [digital image!], JE-00003246, JE-00003247 [digital image!], LY-0084391 [digital image!], WAG-0249595 [digital image!], MO-357730 [digital image!]).=*F. intermedia* var. *annua* Caball. in Trab. Mus. Ci. Nat., Ser. Bot. 30: 30. 1935. *Type*: Morocco. [Western Sahara]: In collibus arenosis insolatis prope Sidi-Ifni, 13 June 1934, *A. Caballero* (**lectotype, designated here**: MA-78660!; isolectotype: MPU-300233!).-*F. pulverulenta* subsp. *floribunda* sensu Quézel & Santa, Nouv. Fl. Algérie: 685. 1963 [sphalm.]. Note: The subspecific epithet “*floribunda*” is most likely a mistaken desinence for “*florida*”, not a formally proposed name.

*Description*: *Habit:* Annual or short-lived perennial herbs, weakly lignified at the base, loosely branched, tap-rooted, and mostly glabrous to sparsely puberulous. *Stems:* Non-rooting, usually prostrate to decumbent, 5–30 cm long, often with divaricate branches, with internodes up to 2.5 cm long. *Young branchlets:* Yellowish to reddish, mostly with scattered whitish depositions, glabrous or shortly and densely puberulous, with minute curled or hooked trichomes (0.1–0.2 mm long) only on one side and near the nodes. *Leaves:* Opposite, patent to erect–patent, green or sometimes reddish, and mostly with scattered glands bearing whitish depositions. *Petiole:* 0.5–0.8 × 0.2–0.3 mm, flattened, and tapering distally. *Sheath:* Extending along the petiole margins to the blade, loosely ciliate to subglabrous, with (2–)4–5 pairs of lateral cilia, unequal in length (0.2–0.5 mm long), whitish, cylindric to flattened, and obtuse or acute at the apex. *Leaf blade:* (1.8–)2.5–4 × 0.5–0.7 mm, triangular–ovate to oblong–ovate, mostly subfalcate upwards, with an obtuse apex and a cordate to rounded base, often strongly revolute on margins or flattened in the lower third, somewhat fleshy and glaucescent, concolorous or sometimes slightly paler abaxially, glabrous on the upper side, loosely hairy beneath with short straight trichomes 0.1–0.2 mm long. *Midrib:* Notably thickened, tapering slightly towards the apex, continuous with the petiole and raised abaxially and extending for most of the blade length. *Flowers:* Pentamerous, perfect, sessile, solitary on dichotomies or borne in axillary or terminal dichasial groups, usually condensed and glomerular. *Floral bracts:* Two, leaf-like but smaller, 2–2.5 × 0.5–0.7 mm, erect to erect–patent, connate at the base and shortly covering the calyx base for ca. 0.5 mm, about half the calyx length. *Bracteoles:* Two, bract-like but smaller, 0.5–1.5 × 0.4–0.6 mm, alternating with bracts, erect, strongly revolute on margins, about half the calyx length, with a petiole ca. 0.5 mm long, adnate to the calyx base. *Calyx:* 2.5–4.2 × 0.6–1 mm, tubular at anthesis to gradually fusiform later, often twisted, straight, indurate, sessile, reddish or rarely yellowish, prominently 5-ribbed, entirely glabrous or densely papillate (with a heterogeneous whitish indumentum of long flattened papillae 0.2–0.3 mm long, globose–claviform papillae ca. 0.1 mm long, and minute globose vesicles), only between the glabrous thickened ribs and sometimes with scattered whitish depositions. *Teeth:* Five, 0.5–1 mm long, erect, triangular, acute, cucullate, shortly mucronate at the apex (mucro ca. 0.1–0.2 mm long), narrowly membranous and shortly papillate at the margins. *Petals:* Five, 4–6 × 1–1.5 mm long, exceeding the calyx, obovate–cuneate, pinkish-mauve but whitish below, slightly overlapping laterally, exceeding about half to two-thirds the length of the calyx. *Claw:* 2–2.5 × 0.3–0.4 mm, narrowly cuneate, tapering to the blade, yellowish, and partly hidden into the calyx tube. *Ligule:* 1.5–2 × 0.2–0.3 mm, narrowly oblanceolate, longitudinally adnate to the claw, the free apex ca. 0.2–0.4 mm long, ovate–oblong, obtuse to subacute, and entire. *Blade:* 2.3–3.5 × 1–1.5 mm, broadly obovate–cuneate, rounded at the apex, irregularly erose–denticulate. *Stamens:* Six, in two unequal whorls, overtopping by ca. 1.5 mm the calyx teeth at anthesis; filaments 3.5–5.5 mm long, whitish, expanded ca. 0.5 mm wide in the lower part but gradually tapering and filiform in the distal part. *Anthers:* 0.6–0.8 mm long, yellowish, ellipsoid, versatile. *Ovary:* ca. 1.5 mm long, ellipsoid, subtrigonous, 3-carpelar. *Placentas:* Three, parietal–basal, extending up to the lower half to two-thirds of the carpel wall length, with the ventral traces moderately to profusely branched. *Ovules:* 9–12 per placenta, attached along most of the placenta by erect funiculi ca. 0.1 mm long. *Style:* 3–4 mm long, terete, whitish. *Stylar branches*: Three, filiform 0.5–0.8 mm long, whitish. *Capsule:* 1.4–2 × 0.8–1 mm, ovoid–ellipsoid, subtrigonous, hidden in the calyx tube, early dehiscent. *Seeds:* 24–30 per capsule, 0.5–0.8 × 0.2–0.3 mm, sulcate on one side, ellipsoid, brown, darker at the funicular part and developing rapidly even before the flower has completely withered. *Testa:* Thin, not mucilaginous, with a surface weakly and irregularly ornamented with a subrectangular–reticulate pattern, finely striate, covered with small-sized papillae 8–17 µm long, homogeneous in shape, globose to conical–obtuse, sessile, more densely disposed on the distal part.

*Etymology*: The specific epithet (*sahariensis*, *–e*) refers to the Sahara Desert, the native area of the species. The name replaces the illegitimate “*F. florida* L.Chevall.”, a later homonym of the Chilean “*F. florida* Phil.” (see below). The type of the former was collected in 1902 and distributed by Louis P.D. Chevallier with number 404 of his exsiccata *Plantae Saharae Algeriensis*.

*Flower and fruit production*: Flowering in early January–early June (occasionally in August–September) and fruiting in February–July (occasionally in September–October).

*Habitat and distribution*: *Frankenia sahariensis* is found in mostly sandy soils and in subsaline depressions, often along ravines and wadies, at elevations of 0–1300 m. It is endemic to the western parts of the Sahara Desert [55], south of the High Atlas Range, spreading from the Atlantic coast of northern Mauritania and southern Morocco to central and eastern Algeria and northwestern Mali (Figure 2). Biogeographically, and according to the current data, it is endemic to the western parts of the Saharan province of the Saharo–Arabian region of Takhtajan [34]. In this vast territory, *F. sahariensis* occurs in halophilous vegetation types within the Oceanic Saharan, Western Saharan, Northwestern Saharan, and Steppe–Northern African domains of Quézel [33].

*Nomenclatural remarks*: Chevallier [56] first described this remarkable plant after his collections near El Goléa (currently El Menia) in the Saharan region of southern Algeria. It was named “*F. florida*”, most probably due to the outstanding showy flowers it exhibits at anthesis. However, the earlier homonym *F. florida* Phil. had already been used for a Chilean congener [57], a fact that made Chevallier’s name illegitimate (Art. 53.1 of the ICN) and unavailable for use. Nonetheless, since the Saharan plant somewhat resembles *F. pulverulenta* (see below), it has often been subordinated to the latter at different infraspecific ranks. First, Maire [58] accepted it as a variety and indirectly validated the name *F. pulverulenta* var. *florida* Maire (Art. 58.1 of the ICN). Later, this basionym was treated by Maire [24] at the subspecies rank, a currently broadly accepted combination (see [4,59]). Nonetheless, when treated at a specific rank, no previous valid names are available for the Saharan plant, and therefore a replacement name, *F. sahariensis* nom. nov. (Art. 6.11 of the ICN), is proposed here for the illegitimate *F. florida* L.Chevall.

*Other studied material*: Algeria (Dza). **Béni Abbès Province:** Sahara, Moyenne vallée de la Saoura, June 1941, *Volkonsky* (MPU-300246!). Sahara: Embouchure de la Saoura, May 1926, *Balachowsky* (MPU-300247!). **Djanet Province:** Sahara Central, Tassili N’Ajjer, lieux humides à Ahrème, en aval d’Ihérir, 7 January 1966, *?* (P-05145224!). **El Menia Province:** Loc d’El Goléa [currently El Menia], vases salées, April 1954, *P. Quézel* (MPU-300243!). **Naâma Province:** Sfissifa, 3 June 1887, *A. Bousquet* (MPU-048475!). Aïn-Sefra, 21 April 1888, *E. Bonnet & P. Maury* (P-06801050!). **Tindouf Province:** Sahara Occidentale: Oued Jehach entre Tindouf et l’Oued Drâa, lieux sableux, printemps 1938, *Ollivier 187* (MPU-048470!). Sahara Occidentale, Tindouf, assez abondant, March 1928, *Dr. Estival* (MPU-048483!). Sahara Occidentale, Region de Chenachane, Chenachane, ? 1923, *Dr. Tripeau* (MPU-300238!). Sahara Occidentale, Nkheila, jardin, 29 March 1995, *Monod 19676* (P-05038753!). Ibidem, zone d’épandage, *Monod 19678* (P-05038755!). Ibidem, *Monod 19679* (P-05038747!). Sahara Occidentale, Entre Nkheila et Rabouni, 30 March 1995, *Monod 19703* (P-05038750!). Mali. **Tombouctou Region:** El Mzereb (El Hank), dans un terrain sale, 20 April 1954, *J. Sougy 392* (P-00799803!). Mauritania. **Dakhlet Nouadhibou Province:** Ténaloul (Ten Alloui), sebkhas du littorale, 22 April 1938, *Murat 2390* (P-05038740!, MPU-300262!). Iouik [Iwik], côte de Mauritania, 9 April 1982, *Th. Monod 18336* (P-05038748!). **Tiris Zemmour Province:** Soudan Français, Chegga-Guelta, 3 January 19 [38], [*O. de*] *Puygaudeau* (P-05038749!). Sahara Occidental, Falaise du Hank, ravin d’Aïn-Chegga, Janvier 1939, *Dr. Rolland 106* (MPU-048484!). Morocco. **Dakhla-Oued Ed Dahab Region:** Dakhla, río de Oro, 28 May 1977, *Th. Monod 16192* (P-05144446!). **Drâa-Tafilalet Region:** Daoura, dans une dava, sur la Hammada de la Daoura, 2 h vers la vallée vers Hachi-Chamba, 9 April ?, *Dr. Le Carbout* (MPU-300240!). Daoura, dans la vallée de la Daoura vers Achi-Chamba, 17 May ?, *Dr. Le Carbout* (MPU-300241!). Entre Ouarzazate et Skoura, 25 March 1932, *F. Peltier* (MPU-048495!). Drâa, graviers de l’Oued Imini à Ouarzazate, 14 February 1936, *Gattefossé* (MPU-967102!). **Guelmim-Oued Noun Region:** Oued Noun, *Ollivier 14* (MPU-300264). In aridis subsalsis inter Labyar et Notfia, 1 April 1937, *R. Maire* (MPU-300242!). Anti-Altas, Goulimine (Guelmin), 23–28 August 1936, *M. Langueron* (MPU-300234!). Prov. Guelmim, 53 km from Guelmin on road to Tan-Tan, near café, 7 February 2007, *S. Jury & T.M. Upson 20492* (MA-786164!). Prov. Guelmime, 14 km WSW of von Guelmime an der Sttraβe (P41) Nach Tan-Tan, ca. 200 m elev., 28°56′ N 10°08′ W, 15 April 1997, *D. Podlech 53647* (W-0365982!, RSA-0548173 [digital image!]). Entre Goulimine (Guelmin) et El Aioun du Drâa, 6 September 1941, *J. de Lepiney, Ch. Runge & Ch. Sauvage 2035* (MPU-759265!). **Laâyoune-Sakia El Hamra Region:** Sahara Occidental: El Aaiun (Seguia el-Hamra), *Izik [Mision d’Études de la Biologie des Acridiens 2482]* (MPU-300239!). Sahara Occidentale: Zemmour [Gueltat-Zemmour], dans les regs près de la Guelta, March-April 1934, *Luthereau* (MPU-300237!). **Souss-Massa Region:** In planitie at septentr. urbis Tiznit, 13 April 1934, *R. Maire & E. Wilczek* (MPU-300236!). Prov. Tiznit, beach west of Gourizime, near the road from Tiznit to Sidi-Ifni, 20 m elev., 29°37′ N 10°02′ W, 14 April 1997, *D. Podlech 53623* (W-0365981!). Prov. Tiznit, Oued Massa ca. 40 km from Tiznit, 50 m elev., 3 April 1994, *A. Tribsch* (W-0365983!). Souss-Massa-Drâa, Tiznit, Ifni Mesti, sur del puerto de Sidi Ifni, 24 March 2015, *D. Gutiérrez* et al. *DG503* (MA-00913227!). Prov. Agadir, 10 km NW of Agadir, at the coast, 120 m elev., 30°29′26″ N 9°40′17″ W, 19 June 1996, *A. Achhal* et al. *96-0511* (W-0365985!). In depressis subsalsis ad Tigert, prope Herculis Promontorium (Cap Ghir), 4 April 1937, *R. Maire* (MPU-300235!). Sud-Ouest du Maroc, Imeoghguemmi, ? 1875, *Mardochée* (MPU-750398!). In arvis prope Tasila, ad ostium fluminis Massa, 3 April 1937, *R. Maire* (MPU-300251!). In arenosis ditionis Tazeroualt, in faucibus Sidi-el-Ghiat, 400 m elev., 12 Abril 1934, *R. Maire & E. Wilczek* (MPU-300263!).

#### 3.1.2. ***Frankenia dinteri*** M.Á.Alonso, M.B.Crespo, Abad-Brotons, Mart.-Azorín & J.L.Villar, ***nom. nov.***

≡*Frankenia densa* Pohnert in Mitt. Bot. Staatssamml. München 1(9–10): 447. 1954. [replaced synonym], *nom. illeg.* [*non* Summerh. in J. Linn. Soc., Bot. 48: 373 (1930)]. **Type** (see Pohnert 1954: 448): Namibia. [Hardap Region: Maltahöhe,] Grootfontein-Süd [MAL 91], 22 November 1934, *K. Dinter 8059* (holotype: M-0104483 [digital image!; available at https://plants.jstor.org/stable/viewer/10.5555/al.ap.specimen.m0104483; accessed on 31 January 2025]; isotypes: M-0104484 [digital image!], PRE-0293175-0 [digital image!], PRE-526699-0 [digital image!] HBG-516900 [digital image!], BOL-136934!, K!). Paratypes: Namibia. [Karas Region:] Bethanien [Bethany], 24 December 1934, *K. Dinter 8270* (BOL-136934!, HBG-516901 [digital image!], K!) (Figure 3).

*Description*: *Habit*: Annual or short-lived perennial herbs, often robust and weakly lignified at the base, profusely and densely branched, tap-rooted, mostly glabrous to sparsely puberulous. *Stems*: Non-rooting, usually erect to ascendant, 5–30 cm long, often with divaricate branches, with internodes up to 2.5 cm long. *Young branchlets*: yellowish-brown to reddish, mostly with scattered whitish depositions, glabrous or shortly and loosely puberulous, with minute curled to hooked trichomes (0.1–0.2 mm long), only on one side. *Leaves*: Opposite, patent to erect–patent, green or sometimes reddish, and mostly with scattered glands bearing whitish depositions. *Petiole*: 0.8–1.2 × 0.5–0.6 mm, flattened, and tapering distally, sometimes reddish. *Sheath*: Extending along the petiole margins to the blade, ciliate with 4–6 pairs of lateral cilia, unequal in length (up to 0.4–0.5 mm long), whitish, cylindric to flattened, and obtuse or acute at the apex. *Leaf blade*: (2.5–)3–4.5(–6) × 0.5–2 mm, narrowly elliptic to oblong, mostly subfalcate upwards, with an obtuse apex and ± a rounded base, often strongly revolute on margins or flattened in the lower third, somewhat fleshy and glaucescent, concolorous or occasionally somewhat paler abaxially, glabrous on the upper side with scattered whitish depositions and waxes, and glabrous or loosely hairy beneath with scattered short straight trichomes 0.1–0.2 mm long. *Midrib*: Narrow, linear, tapering slightly towards the apex, continuous with the petiole below, somewhat raised abaxially, and extending for most of the blade length. *Flowers*: pentamerous, perfect, sessile, often crowded in dense axillary or terminal dichasial groups, usually densely condensed and glomerular, many-flowered. *Floral bracts*: Two, leaf-like, 3–4 × 0.6–0.7 mm, erect to erect–patent, connate at the base and shortly covering the calyx base for ca. 0.5 mm, about equal to or slightly exceeding the calyx length. *Bracteoles*: Two, bract-like but smaller, 1–2 × 0.4–1 mm, alternating with bracts, erect, strongly revolute at the margins, about half to two-thirds the calyx length, with the petiole ca. 1 mm long, adnate to the calyx base. *Calyx*: 3.3–4.7 × 0.6–1 mm, tubular at anthesis to gradually fusiform later, non-twisted, straight, indurate, often reddish, prominently 5-ribbed, entirely glabrous or loosely puberulous (with a homogeneous whitish indumentum of minute straight papillae ca. 0.1 mm long) between the thickened glabrous ribs, often with scattered whitish depositions. *Teeth*: Five, ca. 0.5 mm long, erect, triangular, acute, cucullate, shortly mucronate at the apex (mucro ca. 0.4 mm long), narrowly membranous and shortly papillate at the margins. *Petals*: Five, 3.5–5 × 0.5–0.8 mm, exceeding the calyx, obovate–cuneate, pinkish-mauve but whitish below, often broadly overlapping laterally, exceeding about one-third to half the calyx length. *Claw*: 2–2.5 × 0.3–0.4 mm, narrowly cuneate, tapering to the blade, yellowish, and entirely hidden in the calyx tube. *Ligule*: 2–2.5 × 0.4–0.5 mm, narrowly oblanceolate, longitudinally adnate to the claw, the free apex ca. 1 × 0.5 mm, ovate–oblong, broader than the claw at the base, obtuse to subacute, and entire. *Blade*: 2–2.5 × 0.5–0.8 mm, obovate, rounded at the apex, irregularly erose–denticulate. *Stamens*: Six, in two unequal whorls, hidden in the calyx teeth at anthesis; filaments 2.3–2.5 mm long, whitish, expanded to ca. 0.5 mm wide in the lower part but gradually tapering and filiform in the distal part. *Anthers*: 0.6–0.8 mm long, yellowish, ellipsoid, versatile. *Ovary*: ca. 2 mm long, elliptic–ovoid, subtrigonous, 3-carpelar. *Placentas*: Three, parietal–basal, extending up to the lower half to two-thirds of the carpel wall length, with the ventral traces moderately to profusely branched. *Ovules:* 10–12 per placenta, attached along most of the placenta by erect funiculi ca. 0.1 mm long. *Style*: 1.7–2 mm long, terete, whitish. *Stylar branches:* Three, filiform, 0.5–0.6 mm long, whitish. *Capsule*: 3–4.5 × 1.4–2 mm, ovoid–ellipsoid, subtrigonous, hidden in the calyx tube, early dehiscent. *Seeds:* 25–30 per capsule, 0.5–0.6 × 0.2–0.3 mm, sulcate on one side, ellipsoid, light brown, darker at the funicular part and developing rapidly even before the flower has completely withered. *Testa:* Thin, not mucilaginous, with a surface weakly and irregularly ornamented with a subrectangular–reticulate pattern, finely striate, covered with scattered small-sized papillae 7–12 µm long, homogeneous in shape, conical–obtuse, sessile, more densely disposed on the distal part.

*Eponymy*: The specific epithet honours Moritz Kurt Dinter (1868–1945), a renowned German botanist who extensively collected plants in southwestern Africa, mostly in Namibia; in particular, in 1934, he gathered the specimens upon which Pohnert [60] first described his “*Frankenia densa*”, an illegitimate later homonym of the Australian *F. densa* Summerh. (see below).

*Flower and fruit production*: Flowering in late April–early October (occasionally in November–January) and fruiting in May–November (occasionally in December–February).

*Habitat and distribution*: *Frankenia dinteri* occurs on saline, sandy to clayish soils in wetlands, ravines, or close to lagoons, at elevations of 300–1225 m. The known distribution of the new species extends from Grootfontein Farm (Hardap Region, southern Namibia), the type locality, to Goageb (Karas Region, southern Namibia), but spreads southwards into northwestern South Africa to the Salt River, north of Vanrhynsdorp (Western Cape Province) and far eastwards to Bloemhof (North West Province) (Figure 4). Biogeographically, and according to the current data, it is endemic to the Namib, Namaland, and Karro Provinces of the Karoo–Namib Region (sensu Takhtajan [34]). In that large territory, *F. dinteri* is found in halophilous vegetation-types occurring in the Nama–Karoo (NK), succulent Karoo (SK), and desert (D) biomes [32].

*Nomenclatural remarks*: This outstanding southern African sea-heath was first described by Pohnert [60] as “*Frankenia densa*” from materials collected in inland Namibia, which included robust, profusely branched plants with many-flowered and densely crowded inflorescences. However, the name *F. densa* had already been published by Summerhayes [34] for a sea-heath from central and southern Australia, which made Pohnert’s name illegitimate (Art. 53.1 of the ICN). Apparently, no valid name exists at a specific rank for the southern African plant, and hence a replacement name, *F. dinteri* nom. nov. (Art. 6.11 of the ICN), is proposed here for the illegitimate *F. densa* Pohnert.

*Other studied material*: Namibia. **Karas Region:** Goageb, near the bridge on B4 along the dry river bed, 26°44′54″ S 17°13′13″ E, 922 m elev., 10 August 2016, *M. Mart.-Azorín* et al. (ABH-76804!). Warmbad, im rivierbett, 22 May 1972, *W. Giess & M. Müller 12120* (WAG-0249597 [digital image!]). Seeheim, 17 November 1922, *K. Dinter 4214* (GH-02434737 [digital image!]). Reg. Gross-Namaland, Ganas, 12 December 1884, *H. Schinz 267* (Z-000067508 [digital image!]. Southwest Africa: Kahanstal, 2 December 1934, *K. Dinter 8154* (BOL!, K!). South Africa. **Northern Cape Province:** 2816 (Oranjemund): Sendelingsdrif, stagnant pool along the Orange River near Octha Mine (-BB), 25 August 1982, *W. Metelerkamp 400* (BOL!); Little Namaqualand: Alties River bed, 1 December 1910, *N.S. Pillans 5381* (BOL!); Little Namaqualand: Dry sand near the Kuboos (-BD) stream, October 1926, *N.S. Pillans 5409* (BOL!); Little Namaqualand: Mouth of the Orange River (-CB), on dry mud, lagoon, October 1926, *N.S. Pillans 5576* (BOL!); Little Namaqualand: Mouth of the Orange River (-CB), sandy banks, October 1926, *N.S. Pillans 5583* (BOL!); Namaqualand, Richtersveld Nat. Park, S. Avenants Grove, January 1995, *G. & W. Williamson 5590* (NBG-153968!). 2818 (Warmbad) Regionis occidentalis: Henkries, Orange River (-CC), 30 November 1897, *M. Schlechter 22* (BOL!). 2819 (Ariamsvlei): Daberas Farm (-BD), 28°29′32″ S 19°56′38″ E, 573 m elev., 23 August 2022, *M. Mart.-Azorín*, *M.B. Crespo, M.Á. Alonso*, *J.L. Villar & M. Pinter* (ABH-83264!); Onseepkans, near the Orange River (-CD), 28°46′37″ S 19°15′47″ E, 359 m elev., 21 August 2022, *M. Mart.-Azorín*, *M.B. Crespo*, *M.Á. Alonso*, *J.L. Villar* & *M. Pinter* (ABH-83234!); Kenhardt Distr.: Farm Skroef, bank of the Orange River (-DA), 26 September 1987, *E.V. Hoepen 1887* (BOL!, PRE!). 2820 (Kakamas): Augrabies, (falls) near Park (-CB), 28°39′02″ S 20°21′55″ E, 616 m elev., 23 August 2022 *M. Mart.-Azorín*, *M.B. Crespo*, *M.Á. Alonso*, *J.L. Villar* & *M. Pinter* (ABH-83265!). 2917 (Springbok): Namabeep, dry river bed (-DB), 26 March 1964, *D.S. Hardy 1682* (K!). 3020 (Brandvlei): Between Brandvlei and Williston, Karreekop farm, sandy plains with scattered rocks (-DC), 30°58′59″ S 20°39′18″ E, 987 m elev., 24 August 2022, *M. Mart.-Azorín*, *M.B. Crespo, M.Á. Alonso*, *J.L. Villar* & *M. Pinter* (ABH-83275!). **North West Province**: 2725 (Bloemhof): [Between Christiana and Bloemhof], Vaal River salt works, saltpan (-CB), 27°44′ S 25°20′ E, 29 April 1973, *B. Straschil* (W-0365984). **Western Cape Province**: 3118 (Vanrhynsdorp): 18 miles N of Vanrhynsdorp, Salt River, 22 January 1930, *C.E. Moss 17983* (BM!).

### 3.2. Phylogenetic Relationships

The aligned ITS dataset was 720 bp, 249 (34.58%) of which were potentially parsimony informative. The phylogenetic relationships of the taxa, as recovered in our ML tree, are shown in Figure 5. The percentage of trees in which the associated taxa clustered together (after 10,000 bootstrap replicates) in the ML analysis are shown above the branches, while PP values from the BI analysis are shown below the branches. Trees obtained with the MP and NJ methods (Appendix A) yielded similar topologies to the ML tree. In the MP analysis, the eight most parsimonious trees were obtained with a tree length (TL) of 581 steps, a consistency index (CI) of 0.796, and a retention index (RI) of 0.913.

The general topology of the obtained ITS tree (Figure 5) is almost identical to that reported by Crespo et al. [12], which was based on a more reduced set of species and samples, though it also yields a strongly monophyletic *Frankenia* genus (99BP, 1.00PP) compared with outgroups of the sister family Tamaricaceae (*Tamarix*, *Myricaria*, and *Reaumuria*). The addition of samples of *F. sahariensis* and *F. dinteri*, as well as other congeners from Eurasia and Africa (including Macaronesia), reveals new internal relationships (both within and between clades).

On the one hand, a strongly supported (95BP, 1.00PP) broad aggregate is found that comprises three unresolved major lineages (clades A + B + C) that match those of Crespo et al. [12]. Firstly, Clade A in the ML analysis is composed of two groups of taxa. The first group encompasses all samples of *F. dinteri* from Namibia and South Africa, forming a strongly supported (99BP, 1.00PP) clade that is weakly supported (53BP) as a sister of the remaining groups within Clade A, but it is placed in the BI analysis in an unresolved position together with the broad aggregate of clades (A + (B + C)). The second group is formed by a well-supported (71BP, 1.00PP) broad lineage that includes the South African samples of the strongly supported (96BP, 1.00PP) “*F. repens* group” (sensu Crespo et al. [12]) with *F. anneliseae* M.B.Crespo & al., *F. repens* (P.J.Bergius) Fourc. and *F. nummularia* M.B.Crespo & al.*,* to which the clades of two North African–Iberian species, *F. boissieri* Reuter and *F. thymifolia* Desf., are strongly supported successive sisters. Secondly, samples of the southwestern Mediterranean *F. corymbosa* Desf. (Clade C) are arranged in a strongly supported (99BP, 1.00PP) compact group, which is nested only in the BI analysis as a weakly supported (0.74PP) sister to Clade B. Thirdly, Clade B is formed by two major moderately supported lineages. On one side, Subclade B1 (75BP, 0.78PP) mostly comprises members of the major Macaronesian archipelagos (i.e., the Madeiran *F. cespitosa* Lowe, the Canarian “*F. ericifolia* group” (incl. *F. capitata* Webb & Berthel.), and the Cape Verdean *F. pseudoericifolia* Rivas Mart. & al.) plus two western Saharan species (i.e., *F. ifiniensis* and the newly named *F. sahariensis*), which are arranged into three unresolved lineages. In one of those lineages, all samples of *F. ericifolia* are strongly supported (93BP, 0.99PP) as sister to the *F. sahariensis* clade, and samples of *F. capitata* and *F. ifniensis* Caball. form an internally unresolved but strongly supported clade (99BP, 1.00PP), which is sister to *F. cespitosa*. On the other side, Subclade B2 (99BP, 1.00PP) merges a heterogeneous aggregate of species with the western Mediterranean–Atlantic (*F. laevis*), North African–Atlantic (*F. composita* and *F. velutina* Brouss. ex DC.), or currently subcosmopolitan (*F. pulverulenta*) distributions. Among them, all samples of *F. pulverulenta* form a weakly supported (53BP) clade in the ML analysis.

On the other hand, the “*F. hirsuta* group” (sensu Crespo et al. [12]) consolidates as Clade D, a strongly supported (99BP, 1.00PP) group formed by all samples of *F. hirsuta* s.str. (eastern Mediterranean) plus *F. salsuginea* (central Anatolia), which is sister (59BP, 0.99PP) to the highly supported (95BP, 1.00PP) aggregate of Clades A–C.

Finally, the South African narrow endemic *F. fruticosa* remains basal to all other studied members of *Frankenia*.

The aligned *matK* dataset was 829 bp, 124 (14.95%) of which were potentially parsimony informative. The phylogenetic relationships of the taxa, as recovered in the ML tree, are illustrated in Figure 6. Trees obtained with the MP and NJ methods (Appendix A) yielded similar topologies to the ML tree. In the MP analysis, the nine most parsimonious trees were obtained with a tree length (TL) of 226 steps, a consistency index (CI) of 0.848, and a retention index (RI) of 0.929.

In the obtained *matK* tree (Figure 6), *Frankenia* is also recovered as a strongly monophyletic (100BP, 1.00PP) clade, with two main lineages corresponding to clades ((A + D) + (B + C) of the ITS tree, although they display a different scenario of interspecific relationships. The main clades recovered are partly coincident with those shown in the ITS tree, and their codes are retained in the *matK* tree for parallelism. However, some of those major clades show a lower internal resolution than in the ITS tree. No useful sequences were generated for the South African *F. fruticosa*.

On the one hand, the first lineage is strongly supported (97BP, 0.99PP) and encompasses Clade A (89BP, 0.91PP), with the South African members of the “*F. repens* group” not fully resolved internally, plus Clade D (92BP, 1.00PP), with the “*F. hirsuta* group” including the eastern Mediterranean (*F. hirsuta*) and Anatolian (*F. saslsuginea*) samples.

On the other hand, Clades B + C form a lineage (78BP, 0.97PP) that is sister to Clades A + D and is fully unresolved internally. Firstly, a heterogeneous moderately supported (77BP, 1.00PP) aggregate (Clades B1 + B2) is found that merges species from Clades A, B1, and B2 of the ITS tree. In this group, the northwestern African coastal species *F. velutina* is sister (77BP, 1.00PP) to a clade encompassing the remaining species, which receives weak support (55BP) only in the ML analysis. Provenances of those members are diverse, such as for *F. ericifolia* (Canary Islands), *F. sahariensis* (western Sahara Desert), *F. laevis* and *F. composita* (Mediterranean basin and Atlantic coasts of Northern Africa and Europe), and *F. dinteri* (southern Africa). Although internal relationships among those taxa are not well resolved, three subclades arise in Clades B1 + B2 that respectively include all samples of *F. sahariensis* (64BP, 0.98PP), all samples of *F. laevis* (85BP, 1.00PP), and some samples of *F. ericifolia* (86BP, 1.00PP). Samples of *F. dinteri*, which in the ITS tree formed a strongly supported lineage in Clade A, are unresolved, along with members of Clades B1 and B2. Secondly, the Canarian *F. capitata*, the western Saharan *F. ifniensis*, and the southwestern Mediterranean and North African *F. thymifolia* constitute Clade B1 (60BP, 0.99PP), whose internal relationships are not well resolved. However, an internally unresolved subclade with *F. capitata* plus *F. ifniensis* is found in both of the conducted analyses that is sister (60BP, 0.99PP) to samples of *F. thymifolia*, which, in contrast, nested with members of Clade A in the ITS tree. Thirdly, Clade B2 shows moderate support (62BP, 1.00PP) and encompasses all samples of *F. pulverulenta* (Eurasian–Mediterranean species that became subcosmopolitan) plus the sample of *F. boissieri* (southwestern Mediterranean and African–Atlantic species), without full internal resolution. In contrast, the BI analysis recovered all samples of *F. pulverulenta* in a strongly supported (1.00PP) clade. Members of Clade B2 were included in the homonymous one in the ITS tree, with the exception of *F. boissieri*, which was recovered in the ITS analysis as a member of Clade A. Finally, the southwestern Mediterranean *F. corymbosa* is nested in a strongly supported (96BP, 1.00PP) distinct Clade C, which is recovered as sister (0.75PP) to the former aggregate, but only in the BI analysis.

## 4. Discussion

The findings presented here are largely consistent with those reported by Crespo et al. [12] and form part of a broader study evaluating the generic and specific relationships within Frankeniaceae, with a particular focus on the Eurasian and African taxa.

Both the obtained ITS and *matK* trees (Figure 5 and Figure 6) represent the most comprehensive phylogenies of *Frankenia* to date. While still partial and incomplete, they provide a solid foundation for a preliminary phylogenetic survey on the genus in Eurasia and Africa, which will be expanded in the coming years. When combined with morphological data and field observations, these molecular trees provide a clearer picture, allowing for a more accurate interpretation of certain African taxa that are often synonymised or subordinated to the annual sea-heath *F. pulverulenta* (Table 3). This is exemplified by the two species that have been newly named here, *F. sahariensis* and *F. dinteri*, which are the principal subjects of the present work.

*Frankenia pulverulenta* is native to Eurasia (including the Mediterranean basin) and has become subcosmopolitan after its introduction worldwide [14]. The species is currently distributed in temperate and subtropical regions, ranging from sea level to nearly 2000 m elevation, and it thrives in both very cold regions and hot deserts. According to the taxonomy currently adopted in some general databases [4,26,59], it is the only annual species in the genus, which occurs mostly in regions with low precipitation worldwide and grows optimally in a broad range of soil types with a certain concentration of salts (i.e., chlorides, sulphates, or nitrogen compounds), often in frequented and anthropised areas [61]. For that reason, it is usually regarded as a halo-nitrophilous or gypsum-loving herb [8] with a high degree of morphological plasticity. This lifestyle might be connected to the broad variation it apparently exhibits in terms of body size and indumentum [62]. In addition to its annual cycle, *F. pulverulenta* shows distinctive morphological characteristics that facilitate its identification, including nearly flattened leaves that are cuneate at the base and notched at the apex, solitary scattered flowers, and capsules containing numerous small seeds (Table 3). The presumed morphological polymorphism has led some authors to synonymise *F. pulverulenta* with other congeners with broad, flattened leaves but that strongly differ in many other respects. Recently, Crespo et al. [12] studied two South African species, *F. nodiflora* and *F. nummularia*, long mistakenly identified with the broadly circumscribed *F. pulverulenta*, which widely parallels the present case.

The position of *F. pulverulenta* is not full resolved in our ITS and *matK* phylogenetic trees, although it is consistently placed in a distinct lineage far apart from both *F. sahariensis* and *F. dinteri*. The latter two are not well differentiated molecularly from other relatives, such as the aggregate of *F. laevis* or the aggregate of *F. ericifolia*, but their outstanding morphological divergence allows for their unequivocal identification and strongly suggests recognition at the species rank.

On the one hand, samples of *F. sahariensis* collected in the western Saharan regions are nested far apart from those of the typical *F. pulverulenta* in all the analyses conducted. The position in the ITS tree (Figure 5) of *F. sahariensis* as a strongly supported sister of the Canarian *F. ericifolia* clade is consistent with the geographic distribution of both species, which occurs at about the same latitude in the Atlantic regions of northwestern Africa and the eastern Canary Islands, not far from each other. In contrast, the *matK* tree (Figure 6) includes both latter species in a wide clade with poor internal resolution, which also comprises several morphologically diverse species. Among them, only samples of *F. sahariensis*, *F. laevis*, and, partly, *F. ericifolia* nest individually in well-supported clades, whereas positions of the remaining samples (*F. composita*, *F. velutina*, and *F. dinteri*) are not resolved. All those entities exhibit morphological differences that allow for their easy differentiation at different taxonomic ranks [22]. Conversely, all samples of *F. pulverulenta* (coming from diverse distant places, such as the Mediterranean basin, the Canary Islands, and South Africa) are found together in a heterogeneous weakly supported clade in the ITS tree, which is close to the group containing *F. laevis*, *F. velutina*, and *F. composita*, a group of perennials of mostly Atlantic and southwestern Mediterranean taxa that is sometimes accepted at the subspecific rank in a broadly circumscribed *F. laevis* [22,26]. Furthermore, in the *matK* tree *F. pulverulenta* nests together with *F. boissieri*, but without full internal resolution. Nevertheless, all those phylogenetic relationships are apparently not correlated to sound morphological patterns and can be best interpreted on biogeographical grounds, as discussed below.

In the protologue of *F. sahariensis* (as *F. florida* L.Chevall., *non* Phil.), Chevallier [56] related his new species to the highly variable subcosmopolitan *F. pulverulenta*. In fact, it is often subordinated to the latter as a variety [58] or subspecies [24]. However, the following sound morphological characteristics allow for the confident distinction between both species (Table 3):(i)Leaf blades that are triangular–ovate to oblong–ovate, with a cordate to rounded base and a prominent thickened midrib on the underside of blade in *F. sahariensis* vs. broadly obovate blades with a cuneate base and a thin midrib that is not remarkably prominent in *F. pulverulenta*;(ii)A heterogeneous calyx indumentum, with flattened trichomes between the ribs intermingled with small globose–claviform trichomes and minute papillae in *F. sahariensis* vs. a calyx with a homogeneous indumentum of small curled trichomes also present on the underside of leaves in *F. pulverulenta*;(iii)Petals that exceed half to two-thirds the calyx length (vs. petals exceeding only about one-third to half the calyx length in *F. pulverulenta*); and(iv)Seeds that are morphologically similar in both species (Figure 7), although less abundant in *F. sahariensis* (24–30 per capsule) than in *F. pulverulenta* (up to 45 per capsule).

All these features make individuals of *F. sahariensis* quite distinctive during anthesis in the field. Both species exhibit significant overlap in their distributions and co-occur, at least in the coastal areas, such as in the Ifni region of western Sahara. Caballero [63] described *F. intermedia* var. *annua* Caball. (MA-78660!) from this area, which is now considered a synonym of *F. sahariensis*, as well as *F. pulverulenta* var. *grandifolia* Caball. (MA-78705!), a plant with larger leaves and longer petioles that is indistinguishable from and should be synonymised with the typical *F. pulverulenta*, as suggested by Maire [24] and Nègre [22]. According to Caballero [63], both species occupy distinct ecological niches: *F. sahariensis* (as *F. intermedia* var. *annua*) thrives in abundance in sandy substrates alongside sunny, south-facing slopes, while *F. pulverulenta* (as var. *grandifolia*) is found in scarce quantities in salty and humid substrates along shaded riverbanks.

Chevallier [56] also noted the annual or short-lived perennial habit of *F. sahariensis* (as *F. florida*), stressing its resemblance to *F. laevis* and *F. intermedia* (*F. laevis* var. *intermedia* (DC.) Bonnet), two entities predominantly occurring along the Mediterranean and Atlantic coasts of Europe and North Africa [19]. However, both these latter taxa are woody perennial, shrubby species with prostrate to ascendant stems with denser indumentum all around; their leaves are strongly revolute, not cordate, at the base, and often densely pubescent to tomentose beneath [22]; and the calyx indumentum in both species is always lacking globose–claviform trichomes between the ribs. Furthermore, the calyx ribs in *F. intermedia* exhibit long cilia [64]. Weaker morphological relationships to other relatives occur in northwestern Africa, such as for *F. velutina*, *F. boissieri* or *F. composita* (see Nègre [22]). Nevertheless, some connections can be argued with the Canarian *F. ericifolia*, based on the shared prostrate habit, leaves that are sometimes flattened, and the geographical proximity, which might support the phylogenetic closeness unveiled by the obtained trees (Figure 5 and Figure 6). Despite those similarities, *F. ericifolia* differs significantly from *F. sahariensis* by its shrubby woody habit, the mostly revolute leaves covered with short hairs all over, never being cordate at the base, having a weaker non-thickened midrib, and its homogeneous calyx indumentum with short conical trichomes. For all those reasons, *F. sahariensis* can be readily distinguished from *F. pulverulenta* and its other relatives, therefore deserving recognition at the species rank as previously suggested by Ozenda [55], as *F. florida* L.Chevall.

On the other hand, all samples of *F. dinteri* collected from a substantial territory encompassing southern Namibia and northwestern South Africa are distinctly separate from those of the typical *F. pulverulenta* in all the analyses conducted. Regarding the ITS tree (Figure 5), *F. dinteri* forms a strongly supported group in Clade A that is weakly related to the remaining studied South African species of the “*F. repens* group”. In contrast, the *matK* tree (Figure 6) recovers all samples of *F. dinteri* that do not coalesce into a single clade but that are instead dispersed across Clade D, in which several subshrubby species with diverse origins (*F. sahariensis*, *F. laevis*, *F. ericifolia*, *F. composita*, and *F. velutina*) nest together without complete internal resolution. Once again, phylogenetic relationships of *F. dinteri* appear to respond best to biogeographical patterns than to morphology (see below).

In the protologue of *F. dinteri* (as *F. densa)* [60], Pohnert hypothesised that his newly described species exhibited a close resemblance to *F. laevis*, although it also displayed intermediate characteristics with *F. pulverulenta*. However, *F. laevis* is not known to occur in sub-Saharan Africa, and it shows quite different morphological features, such as the prostrate shrubby stems (vs. annual herbaceous or slightly suffrutescent at the base in *F. dinteri*) with non-fleshy leaves that are strongly revolute and densely pubescent to tomentose beneath (vs. fleshy leaves, often flattened at the base and glabrous to subglabrous beneath in *F. dinteri*), axillary flowers that are scattered along branches (vs. flowers that are often densely crowded in the terminal cymes in *F. dinteri*), and bracteoles about equal to the calyx length (vs. bracteoles about half the calyx length in *F. dinteri*). Furthermore, the original description also presented *F. densa* as a shrublet, though a detailed observation demonstrated that it is a robust annual or short-living perennial herb, only superficially resembling *F. pulverulenta*. On that basis, both names were likely synonymised by Obermeyer [62], a treatment that has been adopted by subsequent authors to date (cf. [65,66]). However, sound morphological differences exist apart from the habit that allow for the easy separation (Table 3) of both latter entities, including the following:(i)Flowers that are often densely crowded in many-flowered cymes in *F. dinteri* vs. flowers that are axillary, solitary, or in lax cymes in *F. pulverulenta* (Figure 3A,B);(ii)Floral bracts that are about half to two-thirds the calyx length in *F. dinteri* vs. those equal to or slightly exceeding the calyx length in *F. pulverulenta*;(iii)Bracteoles 1–2 mm long in *F. dinteri* vs. 2.5–4 mm in *F. pulverulenta* (Figure 3C,D);(iv)Calyx teeth mucro ca. 0.4 mm long in *F. dinteri* vs. ca. 0.1 mm in *F. pulverulenta* (Figure 3D);(v)Anthers that are 0.6–0.8 mm long in *F. dinteri* vs. 0.2–0.4 mm long in *F. pulverulenta*;(vi)Capsule 3–4.5 mm long in *F. dinteri* vs. 2–3 mm in *F. pulverulenta* (Table 3);(vii)Seeds that are morphologically close in both species (Figure 7), although less abundant in *F. dinteri* (25–30 per capsule) than in *F. pulverulenta* (up to 45 per capsule).

As mentioned before for *F. sahariensis*, and according to our data, *F. dinteri* and *F. pulverulenta* overlap in their distributions and are sympatric, at least in the south of its distribution area, i.e., the Karro region of South Africa [12]. They also show ecological behaviours that are analogous to those exhibited by the *F. sahariensis*–*F. pulverulenta* pair in the environs of Sidi Ifni (western Sahara) [63]. In particular, *F. dinteri* thrives abundantly on sandy, drier substrates, whereas *F. pulverulenta* grows in predominantly disturbed, moister soils [63]. However, the distinguishing morphological characteristics that separate each pair remain invariable in habitat and ensure a reliable distinction, which is congruent with their remote position in all our trees (Figure 5 and Figure 6). Resemblances to other congeners are weak and probably due to convergence.

Analogous patterns were documented in other African groups of *Biscutella* L. [67] or the Caryophyllid *Spergularia* (Pers.) J.Presl & C.Presl [68,69]. Incomplete molecular differentiation has been observed in morphologically diverse entities, which is often associated with ecological specialisation. On the one hand, both *F. sahariensis* and *F. dinteri* are native lineages that evolved as local specialists in sandy, rather dry and low-nitrified substrates of ravines and saltpans. On the other, *F. pulverulenta* is a generalist species introduced (alien) to western Sahara and southern Africa that occurs on a broader range of substrate types, often with higher moisture and nitrate contents, particularly in frequented sites. It is only capable of co-existing with the two aforementioned native species on sporadic occasions. This phenomenon is common in the Mediterranean basin and eastern Asia (pers. obs.), where all species of *Frankenia* are native, and has also been reported in similar ecosystems in South Africa [12]. Similar behavioural patterns are expected to be identified in other geographical areas in which *F. pulverulenta* co-occurs with native congeners. Our current data indicate that all three species exhibit distinct ecological behaviours corresponding to diverse stages of specialisation [70], which are connected to precipitation seasonality rather than to evolutionary history (including vicariance), as deduced from our trees. Despite the still incomplete molecular differentiation of *F. sahariensis* and *F. dinteri* in relation to other relatives, the divergent position of both entities in our phylogenetic trees, along with their outstanding morphological differences when compared with the remaining members in the genus, strongly supports their acceptance at a specific rank. Further research, including genetic variation studies of *F. sahariensis* and *F. dinteri* will help to clarify this point.

Finally, in this context, it is worth mentioning the case of *F. cespitosa*, a name that is sometimes synonymised with *F. pulverulenta* [4] or with *F. laevis* L. [26]. *Frankenia cespitosa* was harvested from the islands of Madeira (at Ponta S. Lourenço) and Porto Santo, both from the Archipelago of Madeira (Portugal), and was described by Lowe [71] as a morphologically close relative of *F. ericifolia* and *F. corymbosa*. A thorough examination of the original material (K!, BM!, P!) reveals that the Madeiran *F. cespitosa* is unmistakably a woody perennial species, with leaves that are linear, strongly revolute, and densely covered with whitish salt crusts (as observed on the stems); flowers in dense terminal crowded inflorescences; and a strongly twisted calyx, among other features. All these characters separate the Madeiran plant from both *F. pulverulenta* and *F. laevis*, with the latter producing leaves lacking extensive salt crusts, flowers that are axillary and solitary along the branchlets, and an untwisted calyx. In contrast, the morphological features of *F. cespitosa* closely resemble those of certain species from Macaronesia and northwestern Africa, as suggested in the protologue. In fact, the obtained ITS tree confirms such inferred morphological connections, placing the Madeiran *F. cespitosa* far apart from *F. laevis* as a strongly supported sister to an unresolved clade comprising *F. capitata* Webb & Berthel. (Canary Islands) plus *F. ifniensis* (Atlantic coast of northwestern Africa), which is occasionally treated as *F. corymbosa* var. *ifniensis* (Caball.) Maire, with which it shares gross morphology as well. Furthermore, they are all connected to the *F. ericifolia–F. sahariensis* clade (Canary Islands–northwestern Africa) as well as to *F. pseudoericifolia* (Cape Verde Archipelago), highlighting also the biogeographical relationships between Macaronesia and northwestern Africa. Unfortunately, no *matK* data were obtained to date that illustrate about position of *F. cespitosa* in the plastid phylogeny.

## 5. Conclusions

As currently delineated, many species of *Frankenia* are accepted in such a broad sense that they encompass various biological entities that are superficially similar but are quite diverse when studied in detail. While appearances can be deceptive, the application of integrative taxonomy, combining morphological, phylogenetic, ecological and distributional data, has revealed the existence of distinct biological entities that need to be identified in due measure and that should be accepted into proper taxonomic ranks.

Whilst the results of the present research only inform about the phylogenetic relationships of the Eurasian and African lineages of *Frankenia*, they are sufficient to unravel the taxonomic arrangement (including synonymy) of some critical groups. This is evident in the case of *F. pulverulenta*, which was often circumscribed in a wide sense, encompassing high variability. Accurate investigation of morphological variations alongside thorough analyses of macro- and micromorphological characters is a primary step of paramount importance to achieve a precise identification of taxa in *Frankenia*, which should be accompanied by examining and correctly interpreting nomenclatural types and synonyms. Combining morphology, distributional, and ecologic data with molecular phylogenies often facilitates a more satisfactory re-circumscription of taxa and makes species variation easier to understand.

Accordingly, the morphological and molecular phylogenetic data obtained in the present work show that *F. sahariensis* and *F. dinteri* are different enough to be accepted as specifically distinct from *F. pulverulenta* and other Eurasian and African members of the genus. In short, *F. sahariensis* and *F. dinteri* are native African specialists that thrive on sandy, rather dry and poorly nitrified substrates of inland and coastal subdesert areas, only occasionally co-existing with *F. pulverulenta*. This is a generalist plant introduced into western Saharan and southern African territories, growing in a wider range of soil types that are highly anthropised and often wetter.

Further research is being carried out (i) to expand geographic sampling to fill the current gaps in the distribution areas of the newly named species and (ii) to generate sequences of new plastid regions (i.e., *ycf1*, *trnS-trnG*, and *trnQ-rps16*) to bring more resolution to the phylogenetic trees, and (iii) to add samples of both American and Australian taxa to achieve a more accurate phylogenetic view of the genus across its entire distribution area. The ongoing research on these issues will contribute to complete our understanding of the evolutionary relationships of Eurasian and African aggregates in the global context for *Frankenia*.

## Figures and Tables

**Figure 1 plants-14-01130-f001:**
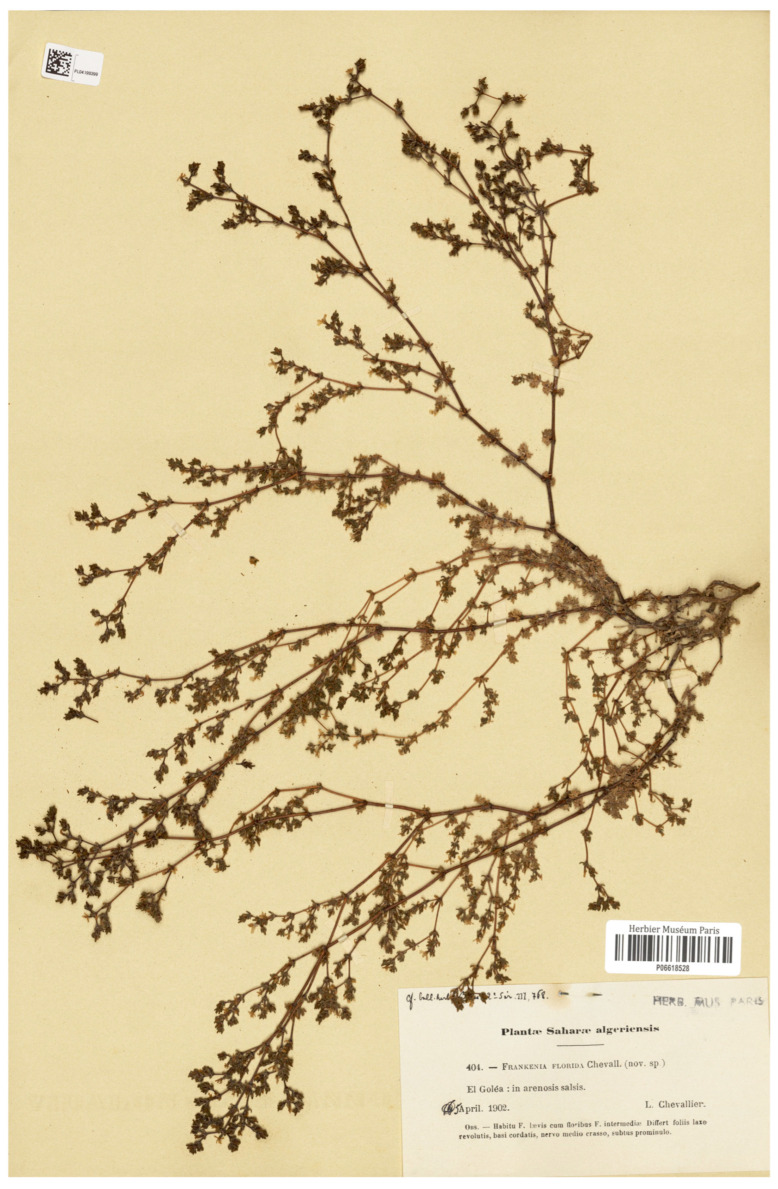
Lectotype of *Frankenia sahariensis* from El Goléa (currently El Menia), Algeria, *L. Chevallier 404* (P-06618528!). Scale bar = 1 cm (on the small upper label). © Muséum National d’Histoire Naturelle, herbarium collections, Paris (reproduced with permission).

**Figure 2 plants-14-01130-f002:**
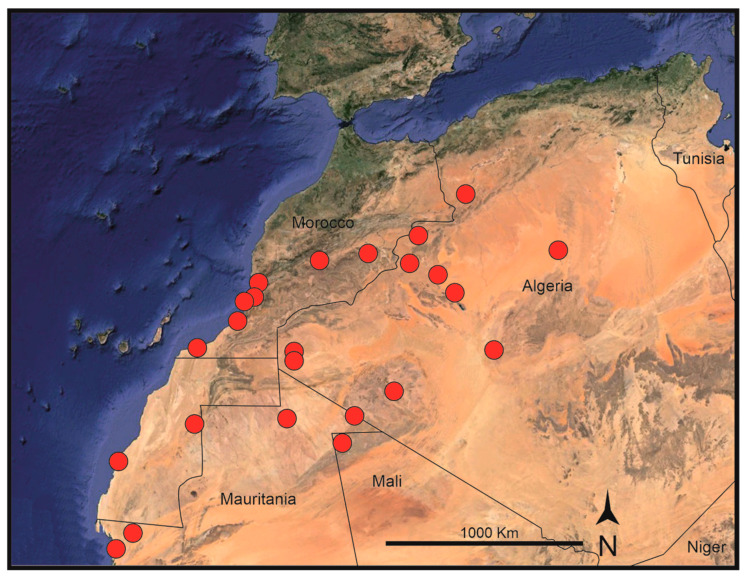
*Frankenia sahariensis.* Distribution map of the studied material.

**Figure 3 plants-14-01130-f003:**
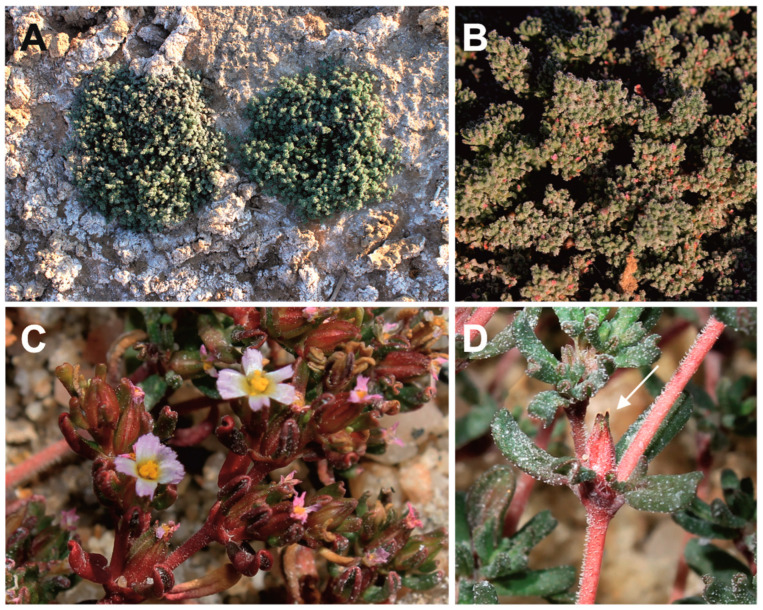
*Frankenia dinteri*. (**A**) Habit in habitat; (**B**) Detail of many-flowered inflorescences; (**C**) Flowering branchlets at anthesis; (**D**) Detail of the calyx after anthesis, with acuminate teeth (arrow). Photographs by Mario Martínez-Azorín: (**A**,**B**) Goageb, Namibia; (**C**,**D**) Onseepkans, South Africa.

**Figure 4 plants-14-01130-f004:**
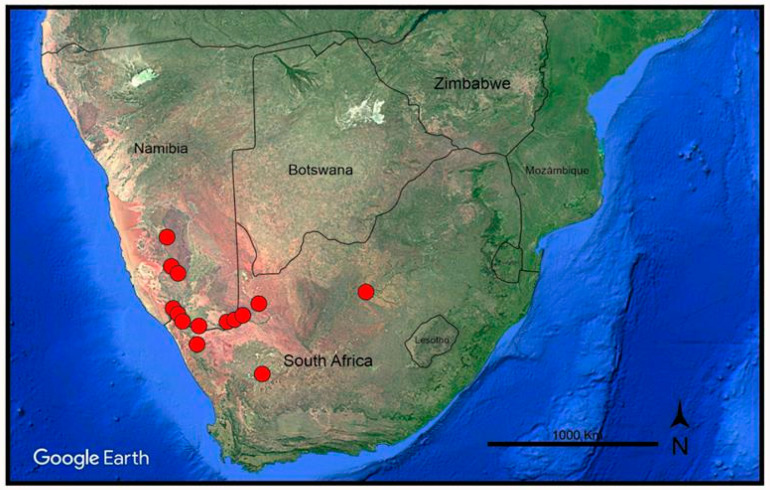
*Frankenia dinteri*. Distribution map of the studied material.

**Figure 5 plants-14-01130-f005:**
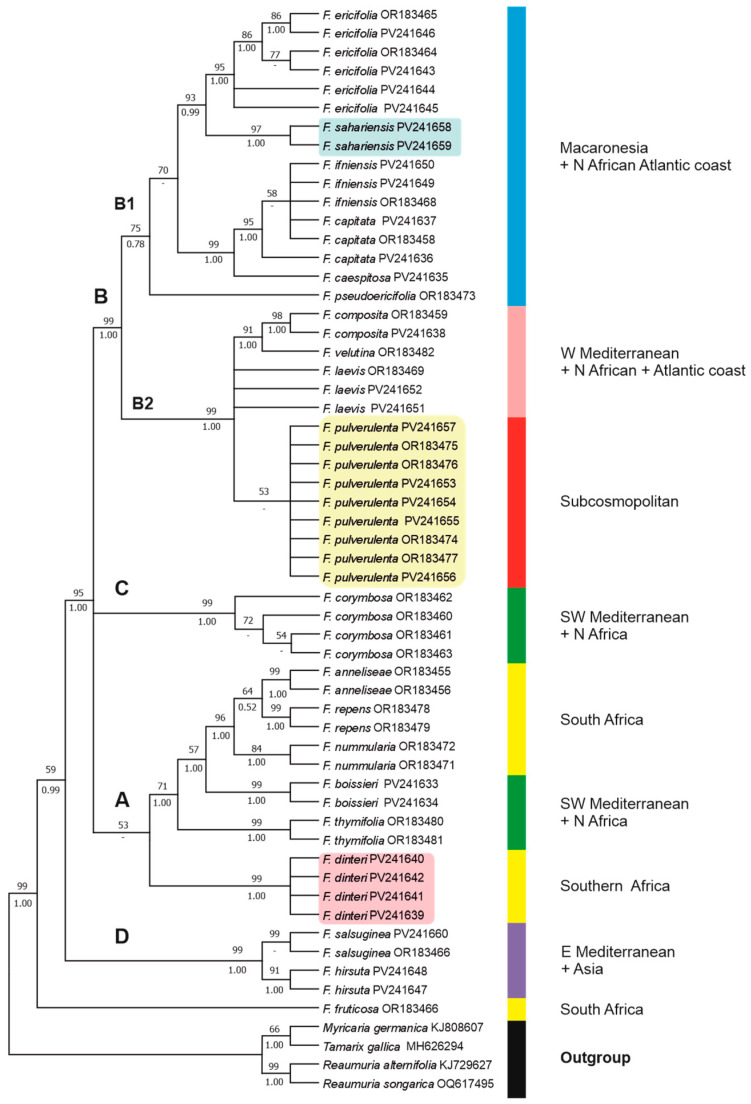
Maximum Likelihood (ML) phylogenetic tree of *Frankenia* accessions from ITS nrDNA sequences as obtained with MEGA. The four main clades recovered in the analyses are marked A–D on the tree, following Crespo et al. [12], plus two additional subclades (B1 and B2). Samples of *F. sahariensis*, *F. pulverulenta*, and *F. dinteri* are highlighted in their corresponding clades or subclades for easier comparison of their relative positions. Numbers above the branches indicate the bootstrap percentage (BP) values after 10,000 replicates from the ML analysis, whereas numbers below the branches indicate the posterior probabilities (PP) from the Bayesian Inference analysis. GenBank codes are shown after each taxon/accession name.

**Figure 6 plants-14-01130-f006:**
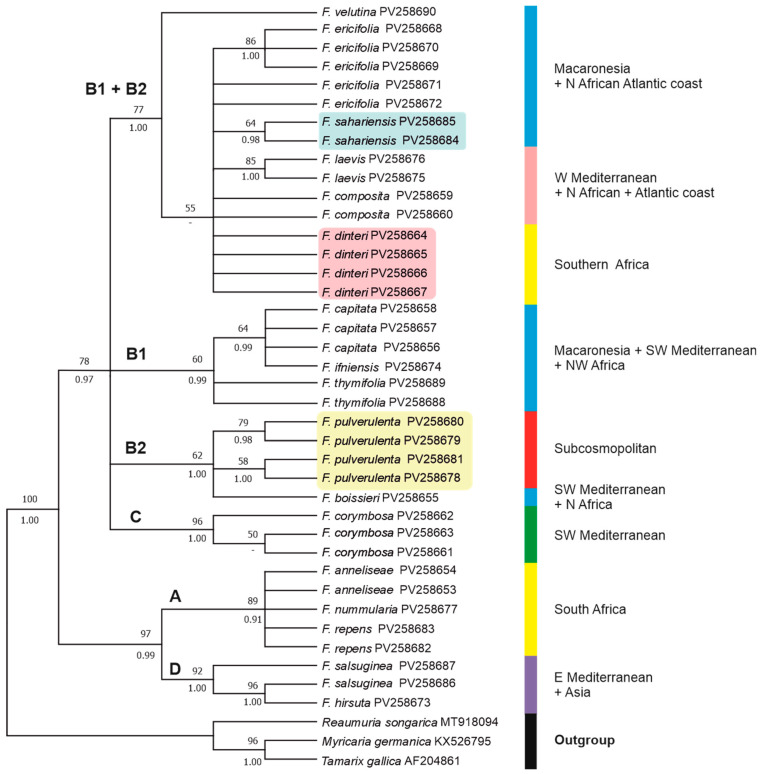
Maximum Likelihood (ML) phylogenetic tree of *Frankenia* accessions from *matK* cpDNA sequences as obtained with MEGA. The four main clades and subclades recovered in the analyses are marked as A–D on the tree, paralleling the ITS tree. Similarly, accessions of *F. sahariensis*, *F.* dinteri, and *F. pulverulenta* are highlighted to easily compare their relative positions. Numbers above the branches indicate the bootstrap percentage (BP) values after 10,000 replicates from the ML analysis, whereas numbers below the branches indicate the posterior probabilities (PP) from the Bayesian Inference analysis. GenBank codes are shown after each taxon/accession name.

**Figure 7 plants-14-01130-f007:**
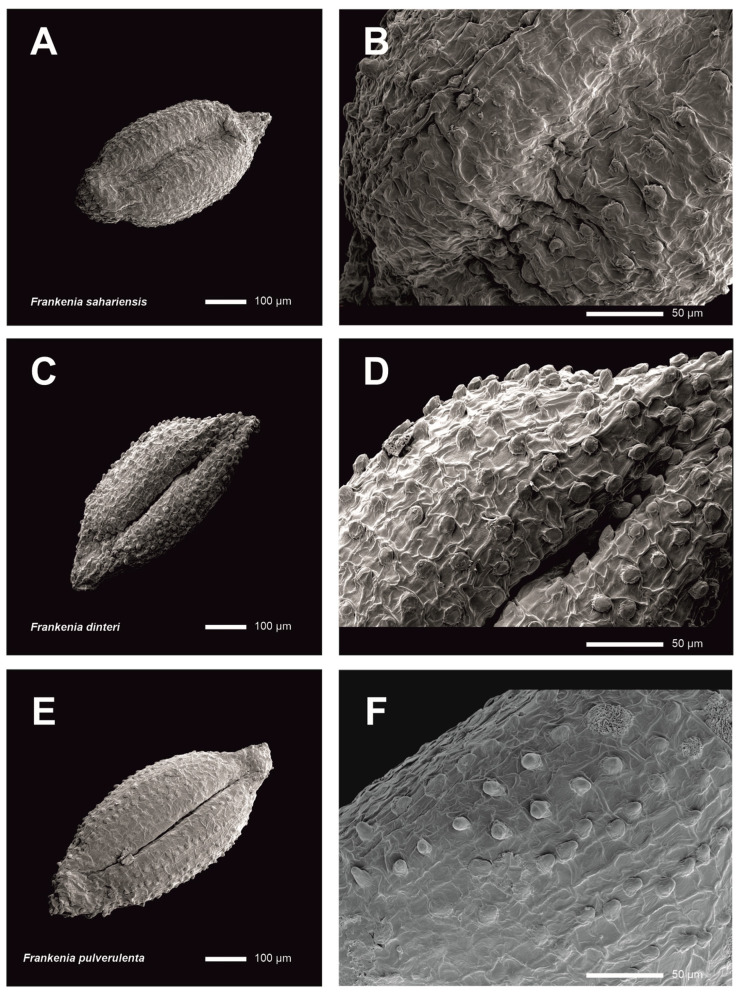
Seed morphology (left) and testa details (right) of (**A**,**B**) *Frankenia sahariensis* (P-05038748); (**C**,**D**) *F. dinteri* (ABH-76891); and (**E**,**F**) *F. pulverulenta* (ABH-7952). Scale bars = 100 µm (**A**,**C**,**E**) or 10 µm (**B**,**D**,**F**).

**Table 1 plants-14-01130-t001:** Seed samples of *Frankenia* for SEM studies, with provenance and herbarium vouchers.

Taxon	Locality	Herbarium Voucher
*F. sahariensis*	Mauritania: Iouik	P-05038748
*F. dinteri*	South Africa: Klipfontein	ABH-76891
*F. pulverulenta*	Spain: Alicante, Orxeta	ABH-7952

**Table 3 plants-14-01130-t003:** Comparison of morphological characters of the newly named African species, *Frankenia sahariensis* and *F. dinteri*, to those of the annual sea-heath *F. pulverulenta*.

	*F. saharienis*	*F. dinteri*	*F. pulverulenta*
General habit	Annual or perennial, herbaceous and weakly lignified at the base	Annual or perennial, herbaceous and weakly lignified at the base	Annual or rarely short-lived perennial
Stem features	Prostrate to decumbent, non-rooting	Erect to ascendant, non-rooting	Prostrate to ascending, non-rooting
Branchlet indumentum	±Densely puberulous on one side and near nodes	±Densely pubescent on one side	±Densely pubescent on one side
Branch trichomes: types and length (mm)	Minute, ca. 0.1–0.2, curled or hooked	Minute, ca. 0.1–0.2, curled or hooked	Minute, ca. 0.1–0.2, curled
Petiole length × width (mm)	0.5−0.8 × 0.2−0.3	0.8−1.2 × 0.5−0.6	0.5–1.5 × 0.2–0.3
Petiole sheath	(2−)4–5 pairs of cilia	4–6 pairs of cilia	2–6 pairs of cilia
Sheath cilia length (mm) and colour	0.2−0.5 mm, unequal, whitish	0.4−0.5 mm, unequal, whitish	0.3–0.6, unequal, whitish
Leaf blade length × width (mm)	(1.8−)2.5–4 × 0.5−0.7	(2.5−)3–4.5(−6) × 0.5−2	2–5 × 1.5–3
Leaf blade outline and colour	Triangular–ovate to oblong–ovate, ±concolorous, midrib notably thickened	Elliptic to oblong, ±concolorous, midrib narrow, not thickened	Obovate–cuneate, ±concolorous, midrib narrow, not thickened
Leaf blade shape and margins	Obtuse apex and cordate to rounded base; often strongly revolute at the margins or flattened in the lower third	Obtuse apex and rounded base; often strongly revolute at the margins or flattened in the lower third	Rounded, slightly emarginate apex and cuneate base; flattened or slightly revolute at the margins
Leaf blade indumentum	Glabrous on the upper side, glabrous to loosely hairy beneath with scattered short straight trichomes 0.1–0.2 mm long	Glabrous on the upper side, glabrous to loosely hairy beneath with scattered short straight trichomes 0.1–0.2 mm long	Glabrous on the upper side, ±densely pubescent beneath with scattered short straight trichomes 0.1–0.2 mm long
Inflorescence	Flowers often in axillary or terminal dichasial groups, usually condensed and glomerular	Flowers in dense axillary or terminal dichasial groups, usually densely condensed and glomerular	Flowers solitary and scattered along branch dichotomies in loose groups
Bracteole length × width (mm)	0.5–1.5 × 0.4–0.6, about half the calyx length	1–2 × 0.4–1 mm, about half the calyx length	2.5–4, as long as or longer than the calyx
Calyx length × width (mm), shape and torsion	2–4 × 0.6–1, fusiform–tubular to fusiform, often twisted	3–4.7 × 0.6−1, fusiform–tubular to fusiform, non-twisted	2.5–4(–5) × 0.8–1.5, fusiform–tubular to fusiform, non-twisted
Calyx indumentum (appearance and length)	Entirely glabrous or densely papillate between the glabrous ribs, with heterogeneous trichomes (flattened papillae 0.2–0.3 mm, globose–claviform papillae ca. 0.1 mm, and minute globose vesicles)	Entirely glabrous or puberulous between the glabrous ribs, with homogeneous papillae ca. 0.1 mm long	Entirely glabrous or puberulous between the glabrous ribs, with homogeneous papillae up to 0.2 mm
Calyx teeth length (mm) and shape	0.5–1, acute, mucronate (mucro ca. 0.1–0.2 mm), cucullate	ca. 0.5, acute, long mucronate (mucro ca. 0.4 mm), cucullate	0.4–0.8, acute to mucronate (mucro ca. 0.1 mm), cucullate
Petal size (mm) and blade colour	5–6 × 1–1.5, pinkish-mauve but whitish below	4–4.5 × 0.5–0.8 pinkish-mauve but whitish below	3.5–5 × 0.6–0.9, whitish-pink to pinkish-mauve, whitish below
Petal blade size (mm), and shape	2.3–3.5 × 1–1.5, obovate–cuneate, rounded and erose–denticulate at the apex	2–2.5 × 0.5–0.8, obovate, rounded to truncate and irregularly erose–denticulate at the apex	2–2.5 × 0.5–0.8, narrowly cuneate to obovate, truncate and erose–denticulate at the apex
Petal claw (mm)	2–2.5 × 0.3–0.4, narrowly cuneate	2–2.5 × 0.3–0.4, narrowly cuneate	2–2.5 × 0.3–0.5, cuneate
Petal ligule (mm)	2–2.5 × 0.3–0.4 mm, free apex ca. 0.2–0.4 × 0.2–0.3 mm, ovate to ovate–oblong, obtuse to subacute, entire	2–2.5 × 0.4–0.5, free apex ca. 1 × 0.5 mm, ovate–oblong, obtuse to subacute, entire	1–2 × 0.2–0.3, free apex ca. 0.4 mm, triangular, acute, entire
Stamen filament length (mm) and morphology	3.5–5.5, expanded ca. 0.5 mm in the lower part, but tapering and filiform in the distal part	2.3–2.5, expanded ca. 0.5 mm in the lower half, but tapering and filiform in the distal part	4–6, expanded ca. 0.2 mm in the lower half, but tapering and filiform in the distal part
Anther length (mm), shape and colour	0.6–0.8, ellipsoid, yellowish	0.6−0.8, ellipsoid, yellowish	0.2–0.4, oblong–ellipsoid, yellowish
Ovules per placenta	9–12	10–12	12–20
Capsule size (mm)	1.4–2 × 0.8–1	3–4.5 × 1.4–2	2–3 × 0.5–1
Seed number and size (mm)	24–30, 0.4–0.5 × 0.2–0.25	25–30, 0.5−0.6 × 0.2–0.3	up to 45, 0.5–0.7 × 0.2–0.3
Testa papillae length (µm)	10–14	7–12	4–17
Papillae morphology and distribution	Homogeneous, globose to conical–obtuse, denser at the distal part	Homogeneous, conical–obtuse, denser at the distal part	Homogeneous, conical–obtuse, denser at the distal part

## Data Availability

DNA sequences generated in the present research are available at GenBank (https://www.ncbi.nlm.nih.gov/genbank/; accessed on 3 April 2025). Other information related to this study will be provided discretionally upon request to the corresponding author.

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
