# Peer review of "Appearances Can Be Deceptive: Morphological, Phylogenetic, and Nomenclatural Delineation of Two Newly Named African Species Related to Frankenia pulverulenta (Frankeniaceae)"

_plants, 2025, doi:10.3390/plants14071130_

Round 1

Reviewer 1 Report

Comments and Suggestions for Authors

The paper "Appearances can be deceptive: Morphological, phylogenetic, and nomenclatural delineation of two newly named African species related to Frankenia pulverulenta (Frankeniaceae)" by María Ángeles Alonso et al. presents a comprehensive study with a strong integrative taxonomic approach. It uses thorough phylogenetic and morphological analyses, resolves nomenclatural issues, and makes a broad contribution to understanding Frankenia diversity in Africa. These elements collectively make it a valuable addition to botanical taxonomy.

The weaknesses of the paper are not particularly serious. There is incomplete phylogenetic resolution, because of reliance on limited molecular markers (ITS and matK). For example, in the matK tree, F. dinteri samples are in a polytomy in Clade D without forming a single cohesive clade. The internal relationships in several clades (e.g., Clade B1) are not well resolved. The sampling of 56 accessions across 20 species is substantial, but it may not fully represent the diversity of Frankenia, particularly given the genus's global distribution (80-90 species, with ~40 in Australia alone). The focus on Eurasian and African taxa, while appropriate for the study’s scope, leaves gaps in representation from other regions (e.g., Australia, the Americas), which may bias our understanding of overall patterns in the genus (especially as one sp., F. pulverulenta is subcosmopolitan).

I suggest that the following changes might improve the paper:
(1) Explain more fully the criteria for selecting outgroups (e.g., Tamarix gallica, Reaumuria spp.) - not explicitly justified beyond their availability in GenBank.

(3) a deeper discussion of the results. While ecological data (e.g., habitat preferences) are included, the paper does not thoroughly explore how ecological differences might drive speciation or reinforce the distinctiveness of F. sahariensis and F. dinteri. For instance, the overlap in distribution with F. pulverulenta is mentioned, but potential niche differentiation or competitive interactions are not discussed in depth, which could weaken the ecological argument for species status. In the same vein, the conclusions mention ongoing research on additional plastid regions, but the paper does not elaborate on other potential avenues (e.g., genomic approaches, population genetics, hybridisation studies, or expanded geographic sampling) that could further validate or refine the findings.

There are several minor errors, of which the following are examples:

Page 10: "February-Juny (occasionally in September-October)" Juny = June or July? Correct error
Page 19: "Petal size (mm) and balde colour" in Table 3 should be "Petal size (mm) and blade colour"
Pages 2/4): "nrDNA" (nuclear ribosomal DNA) is introduced without full expansion on first use (Page 2). It’s later expanded correctly on Page 4 ("nuclear ribosomal DNA (nrDNA)"

Author Response

Dear Reviewer 1,

Many thanks for your comments that much improve the new uploaded version of the Ms. Here you have our responses:

Comment 1: "Explain more fully the criteria for selecting outgroups (e.g., Tamarix gallica, Reaumuria spp.) - not explicitly justified beyond their availability in GenBank."

Response 1: Thanks for this comment. Yes, we agree. In the Mat & Met section (p. 4, lines 194-196), we have added the literature justifying sister position of Tamaricaceae and Frankeniaceae. Further comments in the text seem to be unnecessary at this point, since in the cited literature more details are mentioned about the matter. We hope the new text fixes the gap.

Comment 2: "a deeper discussion of the results. While ecological data (e.g., habitat preferences) are included, the paper does not thoroughly explore how ecological differences might drive speciation or reinforce the distinctiveness of F. sahariensis and F. dinteri. For instance, the overlap in distribution with F. pulverulenta is mentioned, but potential niche differentiation or competitive interactions are not discussed in depth, which could weaken the ecological argument for species status. In the same vein, the conclusions mention ongoing research on additional plastid regions, but the paper does not elaborate on other potential avenues (e.g., genomic approaches, population genetics, hybridisation studies, or expanded geographic sampling) that could further validate or refine the findings."

Response 2: We agree with most of your arguments. To strength the issue of ecological factors and their impact on speciation processes in this group of Frankenia, more discussion concerning the local distribution of each species is now found in the Discusion section (p. 21, lines 716-719; p. 23, lines 782-787; p. 24, lines 790-810), which also implies addition of several new references in the Ref list. In fact, F. sahariensis and F. dinteri are native specialists thriving on drier, not highly nitrified sandy substrates, whereas F. pulverulenta is an introduced alien plant that grows in more humid, nitrate-rich soils (in a broader range of substrate types). We hope this issue is more understandable in the new version. Also, the reference to our ongoing studies Frankenia by adding the concrete plastid regions is completed in the Conclusion section (p. 25, lines 854-861). All these improve significantly the Ms, and so we are very grateful for that.

Comment 3: "There are several minor errors, of which the following are examples:

Page 10: "February-Juny (occasionally in September-October)" Juny = June or July? Correct error
Page 19: "Petal size (mm) and balde colour" in Table 3 should be "Petal size (mm) and blade colour"
Pages 2/4): "nrDNA" (nuclear ribosomal DNA) is introduced without full expansion on first use (Page 2). It’s later expanded correctly on Page 4 ("nuclear ribosomal DNA (nrDNA)."

Response 3: Many thanks for noting these mistakes. Already rectified accordingly in the new version. We have also corrected other mistakes in some parts of the text, and have added other sections (e.g., Appendix A, with the studied material of F. pulverulenta) to improve the new text.

Best regards

Reviewer 2 Report

Comments and Suggestions for Authors

Abstract

Emphasize the practical implications of morphological variability for accurate species identification in Frankenia.

Briefly mention the reinstatement of two species with new names, emphasizing their distinctiveness from F. pulverulenta.

Introduction:

Explain the complexity of Frankenia's taxonomic classification, focusing on morphological variability and misidentification.

Clearly state the study’s purpose and how it addresses gaps in understanding the evolutionary relationships of Frankenia species.

Condense past taxonomic treatments to avoid overwhelming the reader and focus on how they relate to your research.

Materials and Methods:

Ensure consistency by mentioning the sample size or range of specimens examined for each species or population.

Make the explanation of the South African topographical map referencing system more concise, or refer to a supplementary section.

Results

Break down descriptions of F. sahariensis and F. dinteri into shorter sections (e.g., "Morphological Characteristics," "Flowering and Fruit Production").

Discussion

Provide clearer interpretations of the phylogenetic trees, particularly regarding the separation of F. sahariensis and F. dinteri from F. pulverulenta.

Discuss biogeographical patterns supporting species migration, adaptation, and speciation, particularly in desert and coastal environments.

Provide more detail on why morphological distinctions between F. sahariensis and F. pulverulenta are significant, especially where their distributions overlap.

Conclusion

Address areas with low support in the trees to strengthen conclusions.

Discuss how ecological and distributional factors contribute to species differentiation.

Specify the plastid regions targeted in future research to resolve phylogenetic ambiguities.

Author Response

Dear Reviewer 2,

Many thanks for your detailed comments, which much helped us improving the new version of the Ms. We agree with most of them, and here you have our responses:

Comment 1: "Abstract:  /  Emphasize the practical implications of morphological variability for accurate species identification in Frankenia. /  Briefly mention the reinstatement of two species with new names, emphasizing their distinctiveness from F. pulverulenta."

Response 1: In order to fix the first gap, we have added an additional sentence (“Furthermore, the importance of an accurate description of morphological variation in populations is emphasised for a precise identification of taxa in Frankenia.”) in the end of the abstract of the new version. Your second sentence, regarding the mention to the reinstatement of two species with new names is already mentioned in the Abstract, in which we have specified the broad morphological distinction of F. pulverulenta with regard to the new named species. Including also specific differences among those species would expand excessively the abstract. We hope that our corrections improve the text in the way you suggested.

Comment 2: "Introduction: / Explain the complexity of Frankenia's taxonomic classification, focusing on morphological variability and misidentification. / Clearly state the study’s purpose and how it addresses gaps in understanding the evolutionary relationships of Frankenia species. / Condense past taxonomic treatments to avoid overwhelming the reader and focus on how they relate to your research."

Response 2: Regarding your second comment, we have added a new sentence in the last paragraph of the Introduction (p. 3, lines 128-131) to remark the main aims of our study. About your other suggestions, we feel that the current text sufficiently emphasises and summarises the complexity of Frankenia, and it briefly shows the most important contributions to the genus in different areas. Unfortunately, as most contributions are made in local floras, it is rather complex to make a summary table that can facilitate understanding at first sight. We therefore prefer to present the data in the text, avoiding excessive (and perhaps unnecessary) detail, as is the case in the revised text. We hope you understand our preference and decision.

Comment 3. "Materials and Methods: / Ensure consistency by mentioning the sample size or range of specimens examined for each species or population. / Make the explanation of the South African topographical map referencing system more concise, or refer to a supplementary section."

Response 3: We fully agree with your comments. Both points have been corrected in the revised version. The number of vouchers examined for each species is now reported in the Mat & Met section (p.3, lines 146-148). Also, the explanation of the South African grid system has been shortened (p.4, lines 158-160) and a reference to a brief explanation in earlier papers has been included.

Comment 4: "Results: / Break down descriptions of F. sahariensis and F. dinteri into shorter sections (e.g., "Morphological Characteristics," "Flowering and Fruit Production")." 

Response 4: For each species we have presented the morphological, phenological, distributional, ecological, etc. data in a standard way for similar taxonomic work. However, we have changed one of the headings to “Flowering and Fruit Production”, as you suggested.

Comment 5: "Discussion: / Provide clearer interpretations of the phylogenetic trees, particularly regarding the separation of F. sahariensis and F. dinteri from F. pulverulenta. / Discuss biogeographical patterns supporting species migration, adaptation, and speciation, particularly in desert and coastal environments. / Provide more detail on why morphological distinctions between F. sahariensis and F. pulverulenta are significant, especially where their distributions overlap."

Response 5: For each comment, we have made the necessary corrections and additions to the new version of the text. Some new sentences have been added in the Discussion section (e.g. p. 20, lines 669-674). With regard to your second and third comments, new paragraphs have been added emphasising the morphological distinctions of the three species concerned in the overlapping areas and their links with the ecological behaviour (in similar subdesert African areas) of each of them: See p. 21, lines 718-721; p. 22, lines 779-784; p. 23, lines 793-814). However, we cannot add a more detailed discussion of biogeographic patterns and speciation in this group of species, as this is not the focus of the present paper.

Comment 6: "Conclusion: / Address areas with low support in the trees to strengthen conclusions. / Discuss how ecological and distributional factors contribute to species differentiation. / Specify the plastid regions targeted in future research to resolve phylogenetic ambiguities."

Response 6: Based on the newly added information in the Discussion section, we have also included some new or rephrased paragraphs in the Conclusion (p. 24, lines 845-855, 859-871). In addition, the focus of our ongoing research on Frankenia is specified in the last paragraph of the Conclusion section (lines 864-871). What you mention in your second comment is addressed in some new paragraphs in the Discussion section.

Many thanks indeed for your great help with all these.

Sincerely,

Round 2

Reviewer 2 Report

Comments and Suggestions for Authors

Accept in present form

Author Response

Thank you for your kind review.